# Discovery of a three-proton insertion mechanism in α-molybdenum trioxide leading to enhanced charge storage capacity

Yongjiu Lei[1,5], Wenli Zhao[1,5], Jun Yin [1,2,5], Yinchang Ma [1], Zhiming Zhao[1], Jian Yin[1], Yusuf Khan[1], Mohamed Nejib Hedhili[3], Long Chen[3], Qingxiao Wang[3], Youyou Yuan[3], Xixiang Zhang [1], Osman M. Bakr [1], Omar F. Mohammed [4] & Husam N. Alshareef [1] ✉

The α-molybdenum trioxide has attracted much attention for proton storage owing to its easily modified bilayer structure, fast proton insertion kinetics, and high theoretical specific capacity. However, the fundamental science of the proton insertion mechanism in α-molybdenum trioxide has not been fully understood. Herein, we uncover a three-proton intercalation mechanism in α-molybdenum trioxide using a specially designed phosphoric acid based liquid crystalline electrolyte. The semiconductor-to-metal transition behavior and the expansion of the lattice interlayers of α-molybdenum trioxide after trapping one mole of protons are verified experimentally and theoretically. Further investigation of the morphology of α-molybdenum trioxide indicates its fracture behavior upon the proton intercalation process, which creates diffusion channels for hydronium ions. Notably, the observation of an additional redox behavior at low potential endows α-molybdenum trioxide with an improved specific discharge capacity of 362 mAh g$^{-1}$.

Achieving a safe, clean, renewable energy future is a worldwide challenge, which relies on developing efficient energy storage systems that can store the energy obtained from intermittent renewable sources such as solar, tidal, and wind[1–3]. Rechargeable batteries, such as lithium-ion batteries, have achieved success in powering numerous electronic devices and electric vehicles, due to their high energy density and long cycling life[4]. Unfortunately, the uneven distribution of lithium resources and the lack of available reserves limit its application as a low-cost option for large-scale energy storage[5]. Additionally, the use of flammable electrolytes in lithium-ion batteries is a known safety risk, particularly when assembled in massive battery packs[6]. Aqueous batteries, as an alternative energy storage strategy technology, have attracted much attention because of the advantages of low-cost aqueous electrolytes, intrinsic safety, fast transport kinetics, and environmental friendliness[7–9]. These characteristics make them a promising avenue for sustainable exploration in energy storage, as acknowledged in recent studies[10,11].

Non-metallic charge carriers, especially protons, have recently received significant renewed interest from the battery community[12,13]. The single proton is the smallest and lightest charge carrier among all cations, which contributes to fast transport kinetics and minor structural strain during proton (de)intercalation, leading to promising cycling performance[13–15]. In recent years, many electrode materials with high-rate capability were explored using acidic electrolytes and relying on the proton's tiny size and unique hopping transport through the hydrogen bond network[13,16–18]. For instance, Ji and co-workers

[1]Physical Science and Engineering Division, King Abdullah University of Science and Technology (KAUST), Thuwal 23955-6900, Kingdom of Saudi Arabia. [2]Department of Applied Physics, The Hong Kong Polytechnic University, Kowloon 999077 Hong Kong, PR China. [3]Core Labs, King Abdullah University of Science and Technology (KAUST), Thuwal 23955-6900, Kingdom of Saudi Arabia. [4]Advanced Membranes and Porous Materials Center, KAUST Catalysis Center, Physical Science and Engineering Division, King Abdullah University of Science and Technology, Thuwal 23955-6900, Kingdom of Saudi Arabia. [5]These authors contributed equally: Yongjiu Lei, Wenli Zhao, Jun Yin. ✉e-mail: husam.alshareef@kaust.edu.sa

reported defective Prussian blue analog as a proton cathode material, delivering high-rate performance at 4000 C (50% of 1 C capacity) and excellent cycling stability of 0.73 million cycles[15]. Gogotsi and co-workers demonstrated proton insertion in $Ti_3C_2T_x$ MXene using sulfuric acid electrolytes, with super-fast kinetics (210 F $g^{-1}$ at a scan rate of 10 V $s^{-1}$)[19]. Moreover, metal oxides such as $WO_3 \cdot xH_2O$, $MoO_3$, $H_2W_2O_7$, $H_2Ti_3O_7$, and anatase $TiO_2$ were explored as proton anode materials[20–25].

Recently, α-molybdenum trioxide (α-$MoO_3$) attracted much attention for proton storage owing to its easily modified bilayer structure, fast proton insertion kinetics, and high theoretical-specific capacity[26–30]. Some efforts focused on interlayer space regulation of α-$MoO_3$ using ions and water molecules, where they achieved high specific capacity, but with slow kinetics[27]. More recently, proton pre-intercalated $H_{1.75}MoO_3$ demonstrated a quick Grotthuss mechanism[28]. However, it's worth noting that the specific capacity achieved through this approach was relatively lower in comparison to earlier findings. Yamada and co-workers reported the reversible 2.5 protons (de) intercalation process in α-$MoO_3$ as cathode materials, but it needs an additional driving force for reversible cycling[29]. Moreover, the fundamental science of the proton insertion mechanism in α-$MoO_3$ has not been fully revealed till now. Therefore, we explored the proton transport mechanism in α-$MoO_3$ and developed a strategy to mitigate the material dissolution issue.

Specifically, we have uncovered a three-proton intercalation process in α-$MoO_3$ using a phosphoric acid surfactant lyotropic liquid crystalline (PSL) electrolyte. The semiconductor-to-metal transition behavior and the expansion of interlayer spacing in bulk α-$MoO_3$ material after trapping one mole of protons (named $HMoO_3$) were verified experimentally and theoretically. Further investigation on the morphology of $HMoO_3$ indicated its fracture behavior during the proton insertion and extraction process, which created channels for proton diffusion. The improved conductivity, the expanded interlayer spacing, and the increased ion diffusion channels of $HMoO_3$ explain the exceptional rate capability of α-$MoO_3$ as a proton storage material. Thanks to the discovered three-proton insertion mechanism, α-$MoO_3$

exhibited a reversible high specific capacity of 362 mAh $g^{-1}$ as the anode in proton batteries. Our full battery based on α-$MoO_3$ anode and Prussian blue analog cathode delivered a specific energy density of 57.4 Wh $kg^{-1}$ and a maximum power density of 34.9 kW $kg^{-1}$, as well as a long cycling life of 50,000 cycles.

## Results

### Electrochemical performance of the α-$MoO_3$ in acidic PSL electrolyte

To demonstrate the potential of α-$MoO_3$ for large-scale energy storage, we used commercial micro-sized α-$MoO_3$ as the working electrode. The physical characterization results (Supplementary Figs. 1 and 2) confirmed the intrinsic properties of the as-received α-$MoO_3$. To study the proton intercalation mechanism in α-$MoO_3$, and suppress the material dissolution during cycling, we developed a PSL electrolyte with low water activity. Polyoxyethylene lauryl ether surfactant was dissolved in the phosphoric acid and self-assembled into a lyotropic liquid crystalline mesophase electrolyte (Supplementary Fig. 3). To verify the role of the surfactant in PSL electrolyte, Fourier transforms infrared (FT-IR) spectra of different $H_3PO_4$-based electrolytes were collected (Supplementary Fig. 4). The disappearance of PSL electrolyte peaks at 3250 $cm^{-1}$ and 3380 $cm^{-1}$ strongly correlated with robust hydrogen bonding in free water molecules, indicating a decreased amount of free water in PSL electrolyte after adding the surfactant[30]. Additionally, the extended peaks of OH groups and the acid-acid hydrogen bonds around 2750 $cm^{-1}$ and 1630 $cm^{-1}$, and the blue-shifting of the $\nu_{as}$ P(OH)$_3$ peak around 930 $cm^{-1}$ suggest the acid formed a strong hydrogen-bonding network with water and surfactant molecules, as shown in the illustration of Fig. 1a[30,31]. X-ray diffraction (XRD) analysis revealed that the self-assembled PSL electrolyte exhibits a cubic-like structure, as illustrated in Supplementary Fig. 4, with a substantial $H_3PO_4$/surfactant mole ratio of 45[32]. Therefore, the continued hydrogen-bonding network between micelles enabled the hopping migration of protons in a highly viscous PSL electrolyte (a high proton conductivity of 27 mS $cm^{-1}$ at 20 °C, Supplementary Fig. 4c)[33]. The PSL electrolyte exhibited a wide electrochemical stability

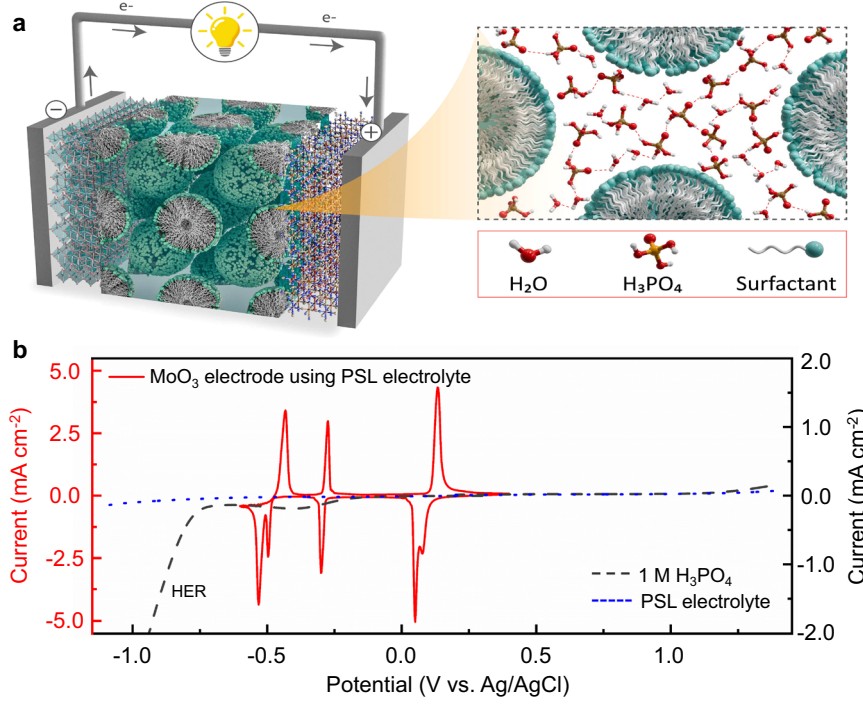

**Fig. 1 | Schematic and stability window of the designed PLS electrolyte. a** Schematic diagram of the PSL electrolyte design. **b** CV curve of α-$MoO_3$ electrode at the scan rate of 1 mV $s^{-1}$ using PSL electrolyte, and the electrochemical stability window of PSL electrolyte using linear sweep voltammetry (LSV) at the scan rate of 0.5 mV $s^{-1}$.

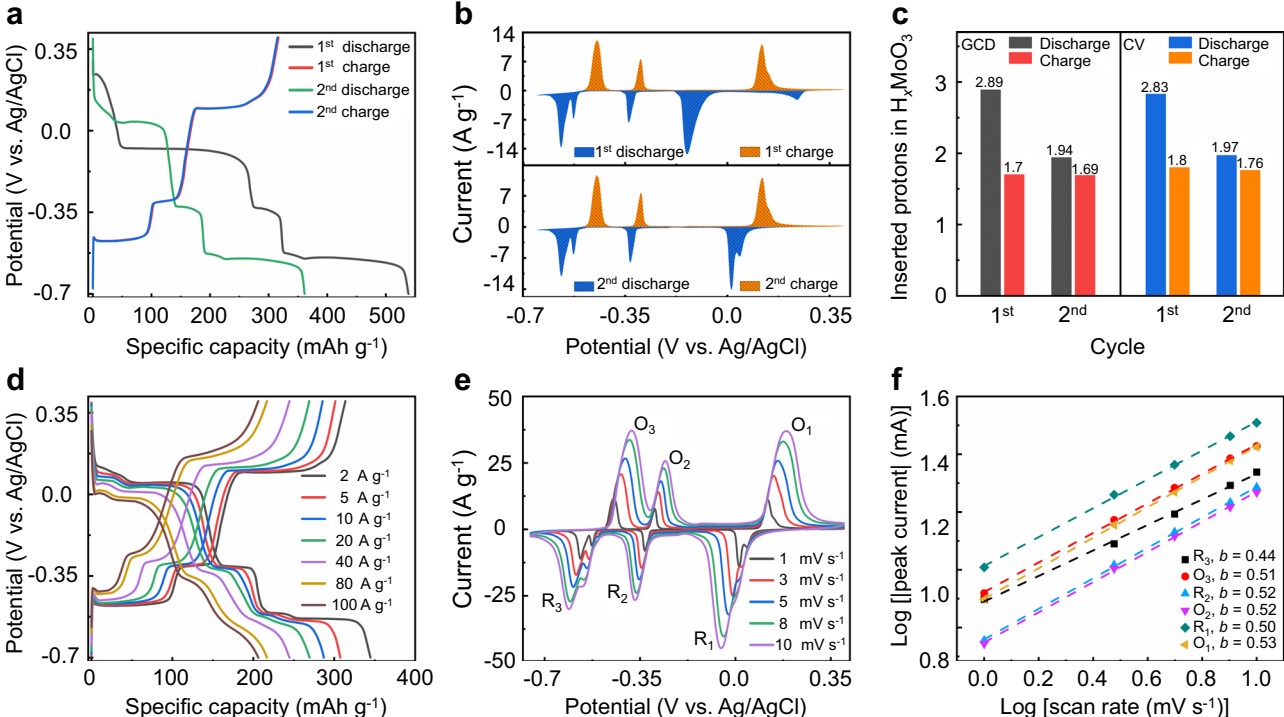

**Fig. 2 | Electrochemical performance of the α-MoO₃ in acidic PSL electrolyte. a** The first and second GCD curves of the α-MoO₃ electrode at 2 A g⁻¹ using PSL electrolyte. **b** The first and second CV curves of the α-MoO₃ electrode at the scan rate of 1 mV s⁻¹ using PSL electrolyte. **c** The inserted protons in H$_x$MoO₃ after the first and second discharge processes of GCD and CV. **d** GCD curves recorded with different rates. **e** CV curves collected at different scan rates. **f** The b values were derived from the CV curves using the equation $I = av^b$.

window and low hydrogen evolution current compared with the 1 M phosphoric acid due to inhibiting the water activity (Fig. 1b)[34].

To explore the electrochemical window of the α-MoO₃ electrode in PSL electrolyte, the cyclic voltammetry (CV) curve (Fig. 1b) was collected employing a Swagelok three-electrode system (Supplementary Fig. 5). Notably, upon scanning to −0.65 V, a previously concealed redox peak of α-MoO₃, typically overshadowed by the hydrogen evolution current, was revealed for the first time. Further elaboration on the proton insertion mechanism underlying this new redox peak will be provided in subsequent discussions. The first galvanostatic discharge (defined as proton (H⁺) intercalation) curve of the α-MoO₃ electrode in PSL electrolyte (Fig. 2a) provided a remarkable specific capacity of 538 mAh g⁻¹ at a current rate of 2 A g⁻¹, which was equivalent to 2.89 mol H⁺ inserted per α-MoO₃ formula unit, closely approaching to the theoretical maximum capacity of 558.6 mAh g⁻¹ (3 mol H⁺ per α-MoO₃ formula unit). However, the first galvanostatic charge (defined as proton deintercalation) process delivered a specific capacity of 315 mAh g⁻¹, equal to 1.7 mol H⁺ was extracted, which indicated that a large amount of H⁺ was likely trapped in α-MoO₃. Consequently, a constant voltage of 0.4 V was applied for three hours following the first charge process to extract the trapped protons (Supplementary Fig. 6). However, this procedure only induced a slight modification of the initial small plateau in the subsequent discharge curve; no capacity change was observed compared to the second discharge cycle in Fig. 2a. This phenomenon highlighted an irreversible reaction mechanism during the first discharge/charge process.

Following this initial cycle, the MoO₃ electrode displayed a reversible discharge capacity of 362 mAh g⁻¹, signifying a 72% improvement in capacity and 230% improvement in specific energy density compared to the two-proton mechanism (Fig. 2a and Supplementary Fig. 7)[35]. To further examine the irreversible insertion behavior, the first and second CV curves of the α-MoO₃ electrode were recorded (Fig. 2b), presenting consistent results with those calculated

from the galvanostatic charge/discharge (GCD) measurements (2.83 mol H⁺ inserted, and 1.8 mol H⁺ extracted in the first cycle). Moreover, the comparison of CV and GCD results is illustrated in Fig. 2c and Supplementary Fig. 8, which confirms the trapping of approximately 1 mol of protons in HMoO₃. This observation aligns well with previously reported values[29].

To study the rate performance of the α-MoO₃ electrode post the initial discharge process, GCD curves were recorded under varying rates (Fig. 2d and Supplementary Fig. 9a), resulting in specific discharge capacities of 354, 307, 285, 227, and 182 mAh g⁻¹ at the corresponding current of 2, 5, 10, 40, and 100 A g⁻¹, respectively. These outcomes suggest a notably enhanced rate capability. The detailed relationship between capacity versus current density is depicted in Supplementary Fig. 10. In contrast, the α-MoO₃ electrode exhibited lower rate capability using 2 M H₂SO₄ electrolyte (Supplementary Fig. 9d), yielding specific discharge capacities of 280, 196, 154, 100, and 76 mAh g⁻¹ at 2, 5, 10, 30, and 60 A g⁻¹, respectively. This attenuation can mainly be attributed to substantial electrode material dissolution[35]. Further insights into the electrochemical performance of the MoO₃ electrodes in both PSL and 2 M H₂SO₄ electrolytes are available in Supplementary Figs. 9 and 11.

In order to comprehend the reaction kinetics of the MoO₃ electrode, the CV curves were obtained at various scan rates using PSL electrolyte (Fig. 2e). The peak current (I) of the three pairs of redox peaks can be correlated to the scan rate (v) through the power-law function, $I = av^b$, where a and b represent variable parameters[36,37]. As presented in Fig. 2f, the b values for O1/R1, O2/R2, and O3/R3 were 0.53/0.5, 0.52/0.52, and 0.51/0.44, respectively, which signifies a diffusion-controlled behavior (b = 0.5). Notably, the R3 reduction peak exhibited a minor shoulder peak at lower scan rates, merging with the main peak as the scan rate increased, mirroring a similar trend observed in the reduction peak R1 (Fig. 2f). These shoulder reduction peaks correspond to the water co-intercalation process, as reported in

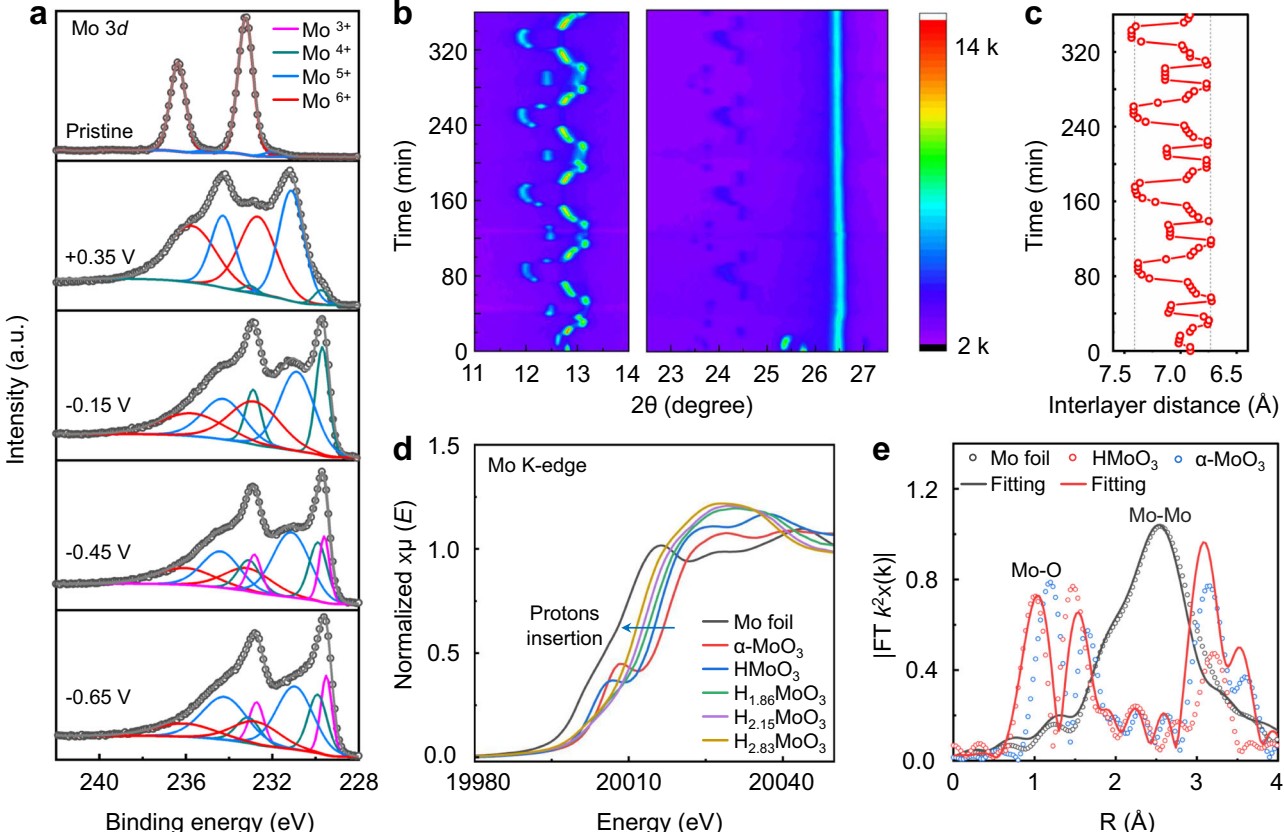

**Fig. 3 | Characterization of the α-MoO₃ structure during proton (de)intercalation. a** The Mo 3*d* XPS spectra of α-MoO₃ electrodes performed at different discharge states. **b** The in-situ XRD curves upon protons insertion and extraction using PSL electrolyte (82 min per cycle). **c** The interlayer spacing of α-MoO₃ derived from the in-situ XRD curves. **d** The XANES spectra of HₓMoO₃ compared to Mo foil. **e** EXAFS spectra and fitting curves of Mo foil and HₓMoO₃ in R space.

previous research, and will be further explored in subsequent investigations[38].

## Characterization and simulation of the α-MoO₃ structure during proton (de)intercalation

Ex-situ X-ray photoelectron spectroscopy (XPS) measurements of MoO₃ electrodes were performed at different discharge states to investigate the mechanism of protons intercalation and confirm the three-proton insertion results calculated from the electrochemical tests. The Mo 3*d* spectrum (Fig. 3a, pristine) exhibits two pairs of 5/2–3/2 spin-orbit doublets. The intense doublet at 233.2 eV and 236.3 eV corresponds to the Mo⁶⁺ oxidation state, while the weaker doublet at 231.2 eV and 234.3 eV corresponds to the Mo⁵⁺ oxidation state[39]. Upon discharging the MoO₃ electrode to 0.35 V after the initial cycle, the presence of Mo⁶⁺ and Mo⁵⁺ oxidation states, along with a third oxidation state (Mo⁴⁺), was observed. This result confirms the trapping of protons in HMoO₃. At lower potentials, further reduction of Mo⁶⁺ and Mo⁵⁺ occurred, leading to an increase in the proportion of the Mo⁴⁺ state, indicating the intercalation of more protons (Fig. 3a, −0.15 V corresponding to sample H₁.₈₆MoO₃). However, in the discharge state at −0.45 V (Corresponding to sample H₂.₁₅MoO₃), two new peaks appeared in the spectrum at 229.3 eV and 232.6 eV, which could be assigned to the Mo³⁺ oxidation state[40]. Upon further reduction of MoO₃ to −0.65 V (Corresponding to sample H₂.₈₃MoO₃), a higher content of Mo³⁺ oxidation state was observed, in line with the expected third proton insertion. These findings provide evidence of the evolution of oxidation states (Mo⁶⁺, Mo⁵⁺, Mo⁴⁺, and Mo³⁺) in the MoO₃ electrode during the discharge process, supporting the progressive intercalation of protons. Furthermore, an examination of the O 1*s* spectrums of MoO₃ electrodes in different discharge states was also

carried out to investigate the protons intercalation process (Supplementary Fig. 12). The O 1*s* spectrum of pristine MoO₃ presented two peaks located at 531.1 eV and 532.4 eV (Supplementary Fig. 12, pristine), which corresponded to lattice O (Mo–O, 94%) and surface contamination (C–O or O=C–O bonds, 6%)[39]. Following proton insertion, the dominant O 1*s* peak of pristine α-MoO₃ split into two peaks, Oₐ at 530.0 eV and O_B at 531.0 eV, indicating the reduction of the Mo⁶⁺ oxidation state[41]. Moreover, an additional O 1*s* peak (designated as O_C) emerged at 532.7 eV, corresponding to the terminal oxygen bonded with protons[39]. The intensity of the O_C peaks demonstrated an escalation during subsequent discharging steps (Supplementary Fig. 12), providing further corroborative evidence for the proposed three-proton insertion mechanism within the MoO₃ electrode.

Considering the limited depth sensitivity of the XPS test on the nanometer scale and the sample size being in the order of tens of microns, further analysis of the valence state of the Mo element in the HₓMoO₃ samples was conducted using Mo K-edge X-ray absorption near-edge structure (XANES) spectroscopy. The XANES spectra clearly indicated a shift towards lower energy absorption near the edge of HₓMoO₃ upon proton intercalation, compared to pristine α-MoO₃, signifying a reduced oxidation state of Mo in the MoO₃ electrodes (Fig. 3d). To further validate the presence of the Mo³⁺ state in the H₂.₁₅MoO₃ and H₂.₈₃MoO₃ samples, we introduced a MoO₂ reference for comparison (Supplementary Fig. 13). The absorption near the edge of H₂.₁₅MoO₃ exhibited a slight shift towards the lower energy region, with a portion of the edge curve remaining at higher energy compared to the MoO₂ reference. This observation confirms the reduced oxidation state of Mo in the H₂.₁₅MoO₃ sample, consistent with the presence of the Mo³⁺ state. In contrast, the absorption near the edge of the H₂.₈₃MoO₃ sample displayed an apparent shift towards the lower

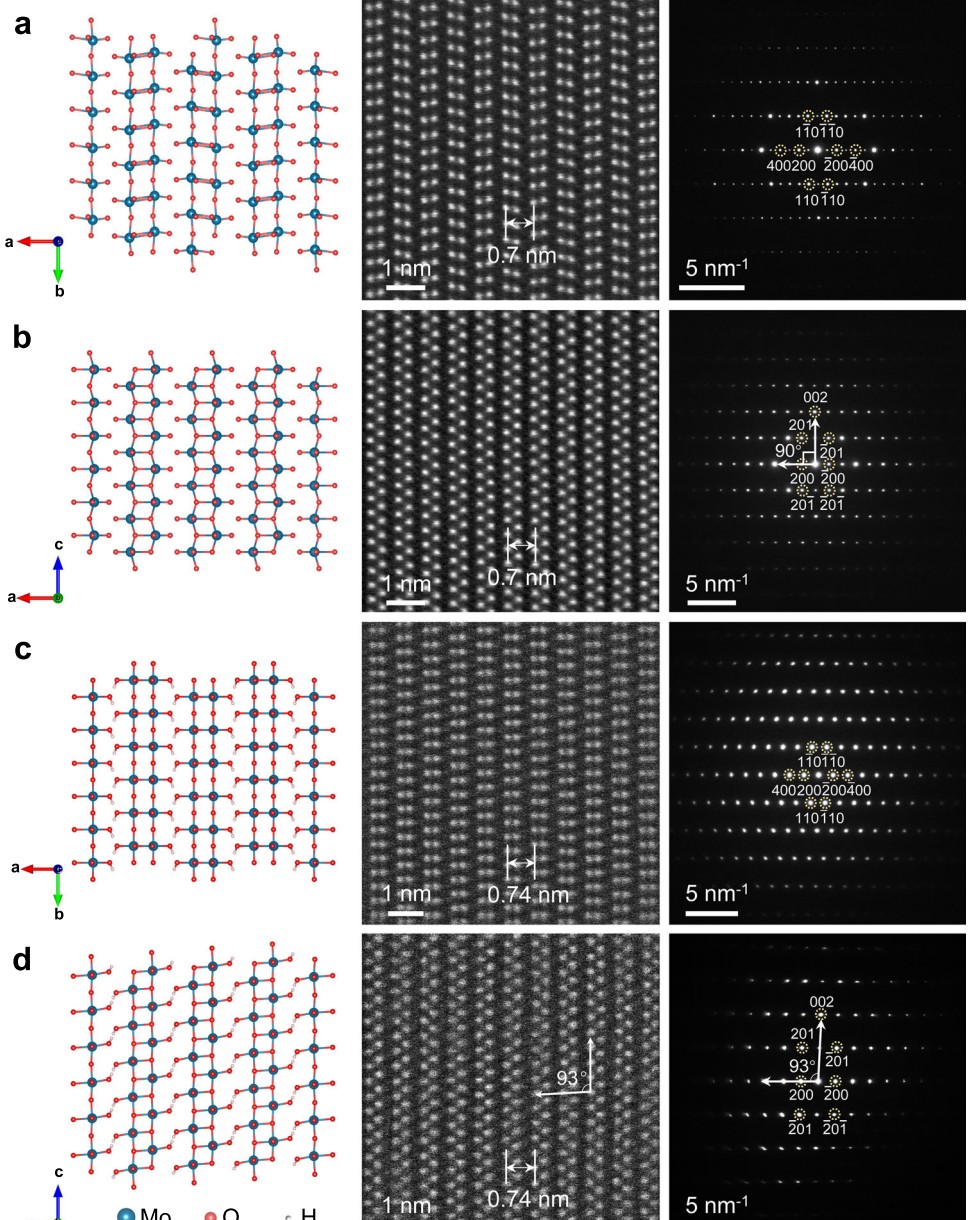

**Fig. 4 | HR-STEM characterization of the α-MoO₃ structure before and after the first cycle. a, b** The crystal structure, HR-STEM, and SAED pattern (from left to right) of pristine α-MoO₃ under different zoom axis (**a**: viewed along the [001] direction and **b**: viewed along the [010] direction). **c, d** The DFT simulated crystal structure, HR-STEM, and SAED pattern of HMoO₃ under different zoom axis (**c** viewed along the [001] direction and **d** viewed along the [010] direction).

energy region, clearly distinguishing it from both the $H_{2.15}MoO_3$ sample and the $MoO_2$ reference. This notable shift confirms a substantial abundance of the $Mo^{3+}$ state in $H_{2.83}MoO_3$, providing further support for the phenomenon of the third proton insertion. Overall, the XANES spectroscopy results not only addressed the limitations of XPS but also provided compelling evidence for the reduced oxidation state of Mo and the presence of the $Mo^{3+}$ state in the $MoO_3$ electrodes upon proton intercalation.

The evolution of atomic structure and crystalline phase of the $MoO_3$ electrodes upon the discharge/charge process was investigated using a combination of low-dose high-resolution scanning transmission electron microscopy (HR-STEM), XRD analysis, and density functional theory (DFT) calculations. The in-situ XRD patterns (Fig. 3b) indicated that the interlayer spacing of $MoO_3$ showed a periodic change with high stability, which was consistent with the reversible charge-discharge curve post the initial discharge/charge process.

During the initial cycle, the pristine α-MoO₃ exhibits an interlayer distance of 6.92 Å, while it decreases to 6.75 Å in the protonation process and increases to 7.38 Å in the deprotonation process (Fig. 3c and Supplementary Fig. 14). This irreversible change in lattice spacing is attributed to the fact that some of the protons were not extracted during deprotonation process, which was also observed in previous reported work using $H_2SO_4$ electrolyte[35].

The lamella samples of $H_xMoO_3$ at different charge states were prepared using focused ion beam (FIB) technique for HR-STEM tests. Figure 4a, b depicts the HR-STEM images of pristine α-MoO₃, highlighting the distinct van der Waals gap (0.7 nm) observed between the zigzag arrays of molybdenum atoms, which aligns very well with the crystal structure (0.693 nm). Also, the corresponding selected area electron diffraction (SAED) patterns in Fig. 4a, b confirmed the orthorhombic symmetry of the pristine α-MoO₃. In Fig. 4c, d, the observed interlayer space expansion (from 0.7 nm to 0.74 nm) and

structural distortion (transformation from orthorhombic to monoclinic crystal structure) by the HR-STEM images clearly indicate the effects of proton intercalation[35]. The corresponding SAED patterns in Fig. 4c, d provide further evidence of the crystal structure distortion, specifically reflected in the change of the $\beta$ angle from 90° to 93°. Notably, these experimental findings are in good agreement with the results obtained from DFT calculations, which yielded a similar $\beta$ angle of 93.7°.

Subsequently, lamella samples of $H_{1.86}MoO_3$, $H_{2.15}MoO_3$, and $H_{2.83}MoO_3$ were prepared using the FIB technique, as depicted in Supplementary Figs. 15–17. Supplementary Fig. 15 illustrates the reduced interlayer spacing in $H_{1.86}MoO_3$ (from 0.74 nm to 0.71 nm when compared with $HMoO_3$) and clearly demonstrates the structural distortion resulting from proton intercalation. The corresponding SAED pattern in Supplementary Fig. 15f provides additional supporting evidence of the crystal structure distortion, manifested by the change in the $\beta$ angle from 93° to 93.5°. Remarkably, these experimental observations are consistent with the results obtained from DFT calculations, which yielded a similar β angle of 93.8°. Similarly, the HR-STEM investigations conducted on $H_{2.15}MoO_3$ and $H_{2.83}MoO_3$ (as shown in Supplementary Figs. 16 and 17) revealed a further reduction in the interlayer spacing (From 0.71 nm to 0.68 nm and 0.69 nm, respectively) and pronounced structural distortion, thus reinforcing the significant influence of proton intercalation. Furthermore, the corresponding SAED patterns in Supplementary Figs. 16f and 17f offer additional evidence of the crystal structure distortion, notably reflected in the changes of the $\beta$ angle from 93.5° to 94° and subsequently to 94.5°.

Moreover, the ex-situ XRD pattern refinement provided valuable structural information and refined lattice parameters of $H_xMoO_3$, as illustrated in Supplementary Figs. 18–22. These findings have been compiled in Supplementary Table 1, demonstrating close agreement with the results obtained from the HR-STEM investigations. Overall, the comprehensive analysis involving HR-STEM, XRD analysis, and DFT calculations provides compelling insights into the evolution of the atomic structure and crystalline phase of the $MoO_3$ electrodes during the discharge/charge process.

The proton-storage sites in the $H_xMoO_3$ crystal structure during the discharge/charge process were investigated using the DFT simulation and extended X-ray absorption fine structure (EXAFS) analysis. Based on the DFT calculations shown in Supplementary Fig. 23, it is revealed that $O_1$ and $O_2$ exhibit potential as proton storage sites, whereas $O_3$ is unsuitable for proton coordination due to its coordination with three molybdenum atoms. The intercalation sequence of protons in $MoO_3$ is as follows: in $HMoO_3$, the most stable state is achieved when protons occupy site 1. Similarly, in $H_2MoO_3$, the most stable state occurs when protons reside at sites 1-1 and 1-2, while sites 2-1 and 2-2 are not coordinated by protons. Lastly, in $H_3MoO_3$, the third proton has the same probability of inserting into site 2-1 or 2-2.

To validate the DFT simulation results, the EXAFS analysis was conducted to gain further insights into the Mo-O coordination (Supplementary Figs. 24–29), and the results are summarized in Supplementary Table 2. According to the theoretical Mo-O distances listed in Supplementary Table 3, the Fourier transform EXAFS (FT-EXAFS) curves of $H_xMoO_3$ exhibit two prominent peaks corresponding to the nearest Mo-$O_1$ and the next nearest Mo-$O_2$ (1.738 Å) coordination, respectively. After the first charge/discharge cycle of the α-$MoO_3$ electrode, the Mo-$O_1$ distance in $HMoO_3$ apparently decreases, while the Mo-$O_2$ distance in $HMoO_3$ slightly decreases (Supplementary Fig. 26) compared to the Mo-O distances in pristine α-$MoO_3$ (Supplementary Fig. 25). These findings confirm the occupation of site 1 by protons in the $HMoO_3$ composite, which aligns with the DFT calculation results. The decreased Mo-O distances also indicate the trapping of protons in $HMoO_3$ after the first cycle, signifying a more stable state

than α-$MoO_3$, which represents an irreversible proton intercalation process.

Subsequent proton intercalation in $H_{1.86}MoO_3$ (Supplementary Fig. 27) results in a slight decrease in the Mo-$O_1$ distance, while the Mo-$O_2$ distance remains the same compared to $HMoO_3$. These observations confirm that protons occupy sites 1-1 and 1-2 in the $H_{1.86}MoO_3$ composite. However, upon further proton intercalation in $H_{2.15}MoO_3$ (Supplementary Fig. 28), the intense peak corresponding to Mo-$O_1$ disappears, and the second main peak (Mo-$O_2$) shifts to a lower distance. This behavior is attributed to the terminal oxygen ($O_1$) being likely to exist in the state of water after combining with two protons, resulting in weak coordination with the Mo atom, ultimately leading to its disappearance. Additionally, the decreased Mo-$O_2$ distance confirms the storage of the third proton on site 2. Further protons intercalation in $H_{2.83}MoO_3$ (Supplementary Fig. 29) leads to a slight decrease in the Mo-$O_2$ distance compared to $H_{2.15}MoO_3$, confirming the storage of the third proton on the bridging oxygen ($O_2$). Overall, the combination of DFT calculations and EXAFS analysis provides valuable insights into the proton storage sites within $MoO_3$ electrodes during the discharge/charge process.

Importantly, the in-situ XRD measurements and HR-STEM observations of $H_xMoO_3$ did not lend support to the water co-intercalation mechanism when using the PSL electrolyte, in contrast to prior research indicating a 14% interlayer space expansion of $MoO_3$ upon water co-insertion when the $H_2SO_4$ electrolyte is using[27]. More efforts involving simulation and in-situ tests of the α-$MoO_3$ electrode have also indicated the absence of water insertion into the crystal lattice of $MoO_3$ upon proton (de)intercalation process[29,35]. Nonetheless, insights from in-situ electrochemical quartz crystal microbalance (EQCM) measurements have suggested the potential occurrence of water co-intercalation within the $MoO_3$ crystal or water adsorption onto the surface of $MoO_3$ particles during the proton (de)intercalation process. This observation was especially pronounced in cases exhibiting a shoulder peak in the CV curve when bulk-size α-$MoO_3$ was utilized as the working electrode material[35,38].

In our study, two analogous shoulder peaks emerged in the CV curve of the $MoO_3$ electrode using the PSL electrolyte (Fig. 1b), which deserved further investigation through EQCM. Regrettably, gel-like electrolytes like PSL are unsuitable for EQCM testing. For comparative measure, we tested the electrochemical performance of the α-$MoO_3$ electrode (Supplementary Fig. 30) in a non-aqueous acid electrolyte (1 M trifluoromethanesulfonic acid in the ionic liquid) under an Argon atmosphere. Interestingly, no discernible shoulder peak was observed in the CV curve, and the third proton insertion phenomenon was obscured by the concurrent hydrogen evolution current. (Supplementary Fig. 30). Therefore, it becomes apparent that water plays a role in the proton insertion reaction within α-$MoO_3$ when tested using the PSL electrolyte, most probably involving water adsorption on the surface of the $MoO_3$ particles, in line with our XRD, STEM measurements and the previous research findings[35]. However, these results have given rise to two important questions: How does this water-involved reaction proceed on the surface of bulk-size $MoO_3$ with its limited specific surface area? And why does the $MoO_3$ electrode manifest a rapid rate capability driven by diffusion-controlled kinetics? Subsequent experiments will be conducted to delve into the underlying mechanisms governing these phenomena.

## Electrical conductivity and structure characterization of the α-$MoO_3$ during proton (de)intercalation

While the correlation between rapid reaction kinetics and the metallic behavior of $MoO_3$ nanobelts has been addressed in previous research findings, more direct evidence is necessary to validate the semiconductor-to-metal transition of bulk-size $MoO_3$ during the process of proton insertion and extraction[28]. To investigate the conductivity variation of $MoO_3$ particles across various discharged states,

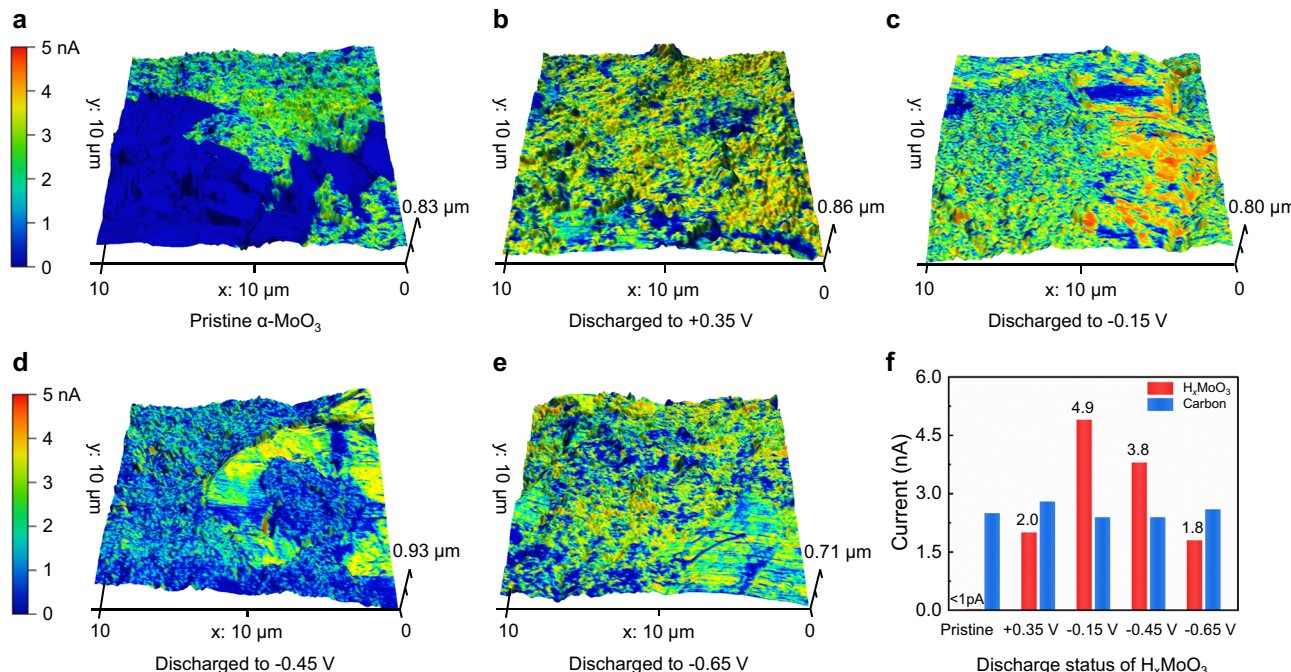

**Fig. 5 | 3D topographical and current mapping of the α-MoO₃ electrode at different discharged states.** (3D topographical and current mapping was the combination of topographical AFM images and 2D current mapping images) **a** pristine α-MoO₃ electrode. **b** The MoO₃ electrode discharged to +0.35 V (HMoO₃). **c** The MoO₃ electrode discharged to −0.15 V (H$_{1.86}$MoO₃). **d** The MoO₃ electrode discharged to −0.45 V (H$_{2.16}$MoO₃). **e** The MoO₃ electrode discharged to −0.65 V (H$_{2.83}$MoO₃). **f** The average current response of additive carbon and the H$_x$MoO₃ particles at different discharged states.

current-sensitive atomic force microscopy (CS-AFM) was employed to measure the current response when a constant voltage was applied (−0.5 V DC bias). As shown in Fig. 5, the 3D topographical and current mapping of the MoO₃ electrode at different discharged states were conducted to estimate the conductivity of the MoO₃ particles and the carbon additive (more details were shown in Supplementary Fig. 31). As expected, the pristine MoO₃ particle presented a low current response (<1 pA) due to its inherent semiconductor nature. However, remarkable current responses of MoO₃ particles were recorded following proton intercalation, indicating comparable or even superior conductivity in comparison with the reference additive carbon. The average current response of MoO₃ particles at different discharged states (Fig. 5f) displayed the conductivity regularity of H$_{1.86}$MoO₃ (discharged to −0.15 V) > H$_{2.15}$MoO₃ (discharged to −0.45 V) > HMoO₃ (discharged to 0.35 V) > H$_{2.83}$MoO₃ (discharged to −0.65 V), suggesting an alteration in electrical conductivity upon proton insertion. Interestingly, the charge transfer resistance (R$_{ct}$) and the overpotential of the CV curves (Supplementary Fig. 32) at different discharged states of MoO₃ electrodes exhibited an inverse trend compared to the current responses. This observation provides evidence for a positive correlation between the conductivity and proton insertion kinetics in MoO₃ particles.

Furthermore, the cracking behavior of MoO₃ particles was observed after the first discharge/charge process (Supplementary Figs. 33–36), which could potentially be attributed to the (de)intercalation of hydronium ions. To gain insight into the structural transformations occurring within MoO₃ particles during the discharge/charge process, a cross-sectional TEM lamella of the MoO₃ particle was prepared using the FIB and subsequently subjected to the low-dose HR-STEM test. As shown in Fig. 6b and Supplementary Fig. 37, numerous linked cracks were observed in the HMoO₃ lamella sample compared with the pristine α-MoO₃ sample (Supplementary Fig. 38). This phenomenon led to the generation of newly interior surfaces within the MoO₃ particle, effectively facilitating the water absorption and proton diffusion throughout the discharge/charge process.

Additionally, the cracking behavior induced fractures in the dense surface oxide layers of the MoO₃ particles (Supplementary Fig. 39), thereby enhancing ion diffusion within the particle structure and subsequently contributing to its rapid rate capability.

To gain further insights into the structural evolution of MoO₃ particles over prolonged, the 20000 GCD cycled MoO₃ electrode was prepared using the PLS electrolyte. Remarkably, the micro-sized MoO₃ particles split into numerous nanobelts (Fig. 6c), a morphology reminiscent of those synthesized via the hydrothermal method (Supplementary Fig. 40)[38]. As the illustration shows in Fig. 6a, this cracking behavior could be attributed to the decreasing van der Waals forces and the hydronium's osmosis over the long cycling process. To accurately characterize the semiconductor-to-metal transition of MoO₃, resistance measurements upon temperature change were carried out with a four-terminal configuration. The 20000 GCD cycled MoO₃ nanobelt (HMoO₃) was patterned on the SiO₂ (300 nm)/Si substrate via electron-beam lithography (Fig. 6d), followed by e-beam evaporation of Ti (10 nm)/Au (90 nm) metals (Supplementary Fig. 41). For comparison, the pristine MoO₃ belts were synthesized through the hydrothermal method, then patterned in the same way (Supplementary Figs. 42 and 43). The resistivity of the cycled HMoO₃ nanobelt showed a linear decrease with decreasing temperature, indicating its metallic behavior (Fig. 6e). In contrast, the resistivity of the pristine α-MoO₃ belt increases exponentially with decreasing temperature, signifying its semiconducting nature. These experimental results are consistent with the DFT calculations (Supplementary Fig. 44) and previously reported findings on H$_{1.68}$MoO₃ prepared through chemical hydrogenation[42].

**Full battery performance of the α-MoO₃ in acidic PSL electrolyte**
To further demonstrate the concept of the aqueous proton battery for practical applications, we assembled the full battery by employing α-MoO₃ as an anode and copper hexacyanoferrate (CuFe-PBA) as the cathode (Fig. 7a). The GCD curves of the full battery were recorded at different current densities using the PSL electrolyte (Fig. 7b), which

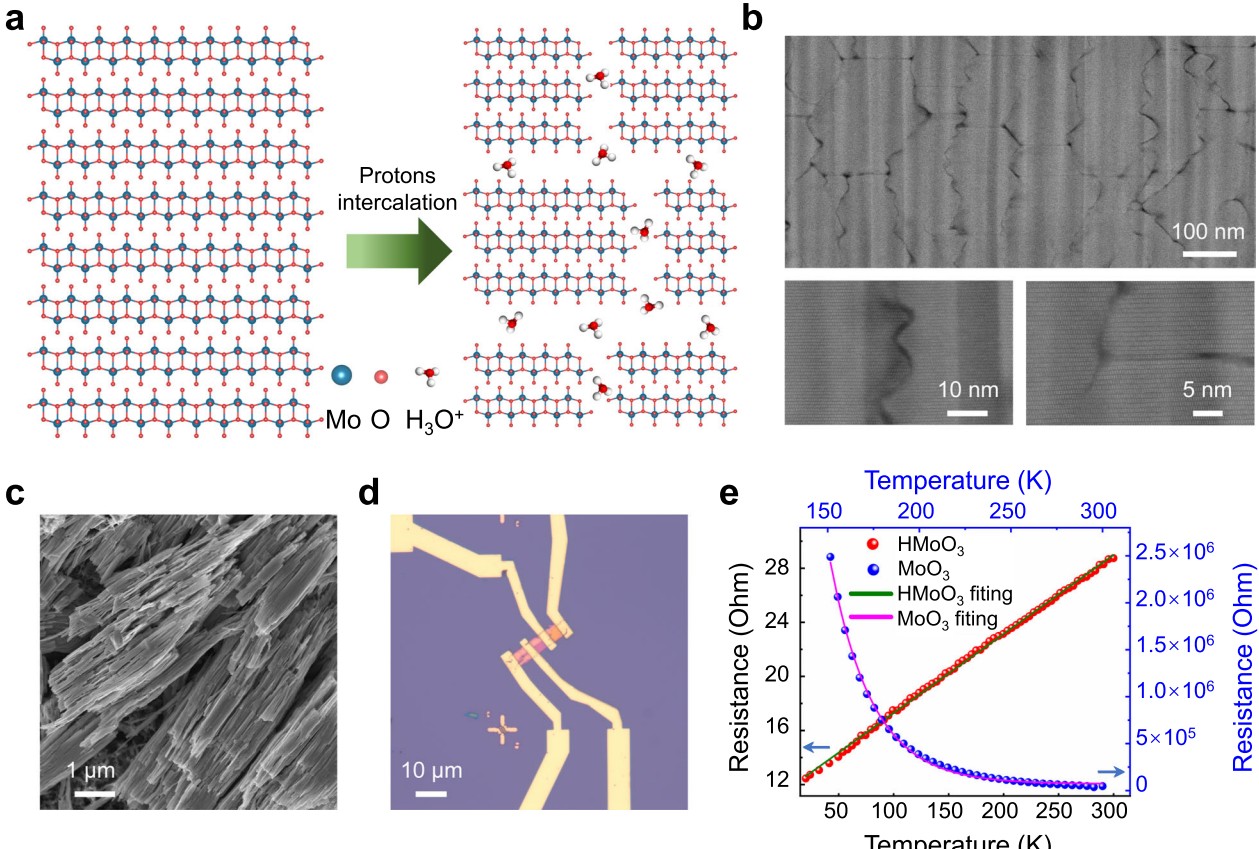

**Fig. 6 | Conductivity characterization of the cycled MoO₃ belt. a** The illustration of cracking behavior in the MoO₃ particles. **b** The HR-STEM images of HMoO₃ particle. **c** The SEM image of micro-sized MoO₃ particles which split into numerous nanobelts upon proton insertion and extraction. **d** Optical photo of the patterned device using the nanobelt split from the bulk MoO₃ particles. **e** Resistance measurements of the pristine α-MoO₃ belt and the cycled HMoO₃ belt upon temperature change.

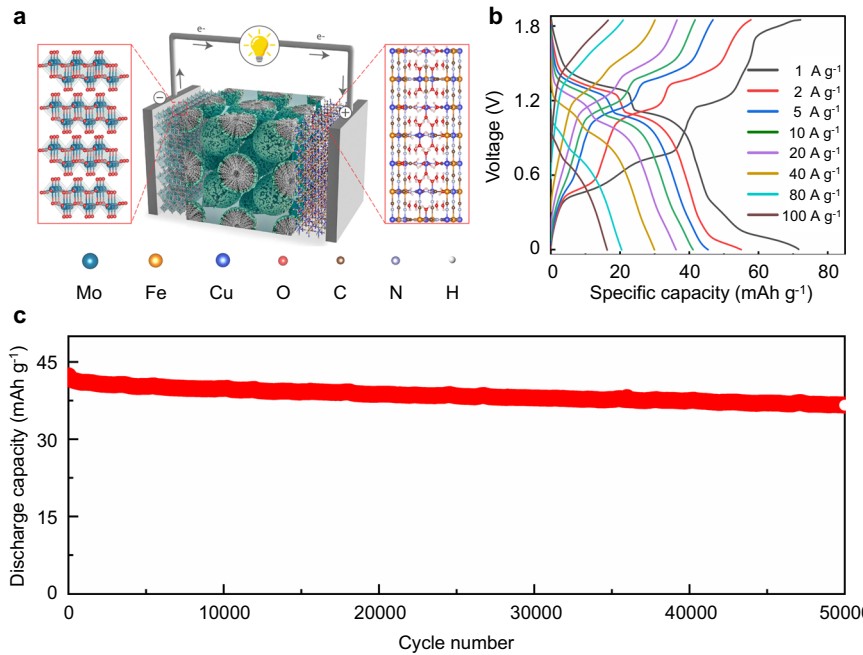

**Fig. 7 | Full battery performance. a** The illustration of the full battery concept using CuFe-PBA as a cathode, designed PSL electrolyte and α-MoO₃ as an anode, and. **b** GCD curves of the full battery recorded at different current densities. **c** Cycling performance of full battery at 10 A g⁻¹ using the designed PSL electrolyte.

delivered the highest discharge capacity of 71.6 mA h g$^{-1}$ (calculated based on the mass of both anode and cathode active materials). The corresponding power and energy densities were shown in Supplementary Fig. 45, delivering a specific energy density of 57.4 Wh kg$^{-1}$ at a power density of 0.56 kW kg$^{-1}$ and a maximum power density of 34.9 kW kg$^{-1}$ at the energy density of 12.6 Wh kg$^{-1}$. Moreover, the capacity was maintained at 16.4 mA h g$^{-1}$ when the current density increased to 100 A g$^{-1}$, indicating a remarkable rate performance (Supplementary Fig. 45). Impressively, the full battery demonstrated promising cycling performance, retaining 87% of its initial capacity after 50000 cycles (Fig. 7c). This result indicates its suitability for applications requiring long cycling life for energy storage.

## Discussion

In summary, we have discovered a new three-proton intercalation mechanism in α-MoO$_3$. The semiconductor-to-metal transition behavior and the proton-trapping process in bulk MoO$_3$ were verified experimentally and theoretically. Further investigation of the morphology of MoO$_3$ indicated its fracture behavior upon the proton (de) intercalation process, which created diffusion channels for hydronium ions. The MoO$_3$ electrode exhibited a reversible high specific discharge capacity of 362 mAh g$^{-1}$ due to the newly revealed redox peak (72% improvement in capacity and 230% improvement in specific energy density compared to the two-proton mechanism). This work provides new insights into the electrochemistry of proton intercalation in α-MoO$_3$ and demonstrates its promising future for high-power and large-scale energy storage applications.

## Methods

### Materials

Micro-size α-MoO$_3$ (≥99.5%) and N-methyl-2-pyrrolidone were purchased from Honeywell. Sulfuric acid (98%), Phosphoric acid (85%), Polyoxyethylene (23) lauryl ether, Trifluoromethanesulfonic acid (HOTf, >99%), and 1-Ethyl-3-MethylImidazolium Bis (Tri-FluoroMethylSulfonyl) Imide (EMITFSI, ≥97%) were purchased from Sigma-Aldrich. Copper sulfate (CuSO$_4$) pentahydrate and potassium ferricyanide (K$_3$Fe(CN)$_6$) were purchased from Fisher Scientific. Conductive carbon Super C65, polyvinylidene fluoride (PVDF, ≥99.5%), and titanium foil (thickness of 50 μm, 99.99%) were purchased from MTI corporation. All these chemicals were used as received. The carbon paper was purchased from Fuel Cells Earth and annealed at 500 °C under Argon before use. The glass-fiber separator was purchased from Whatman. Deionized water was used in all the experiments. The CuFe-PBA particles were synthesized following the reported method with some modifications. Typically, 20 mL CuSO$_4$ solution (0.2 M) was dropped into 20 mL K$_3$Fe(CN)$_6$ (0.1 M) solution under magnetic stirring (800 rpm). After 6 h, the precipitate was washed with DI water three times and collected via a centrifugal force of 2012 × g. Then it was frozen and dried at −50 °C under a vacuum condition.

### Material Characterization

Transmission electron microscopy images were performed by Titan 80-300 ST, FEI. High-resolution scanning transmission electron microscopy images, selected area electron diffraction patterns, and energy dispersive X-ray spectroscopy mapping were performed by Titan Themis Z, FEI. Scanning electron microscopy images were conducted by Merlin, Zeiss, Germany. The lamella samples were prepared by Helios G4 UX dual beam system. The XRD patterns were recorded by a Bruker diffractometer (D8 Advance) with Cu Kα radiation, where the wavelength was 0.15406 nm. Operando XRD experiments were conducted by a Bruker D8 Advance Twin with Cu Kα radiation (λ = 0.15406 nm). Differential Scanning Calorimeter (DSC) measurements were performed from 20 °C to −80 °C and then back to 20 °C at 5 °C min$^{-1}$ (DSC-TA Discovery 250) under a nitrogen atmosphere. Fourier transform infrared (FT-IR) spectra were recorded in the range

of 700 to 4000 cm$^{-1}$ by a Nicolet 6700 spectrometer. The current-sensitive atomic force microscopy (CS-AFM) analysis was performed in Tunneling Current AFM mode on the Dimension Icon Atomic Force Microscope system (Bruker, Santa Barbara, CA). Resistance measurements of MoO$_3$ were carried out with a four-terminal configuration and low-temperature electrical measurements were performed in a Physical Property Measurement System (Quantum Design). X-ray photoelectron spectroscopy (XPS) analyses were conducted using a Kratos Axis Ultra DLD spectrometer outfitted with an Al Kα X-ray source (hv = 1486.6 eV) operating at 75 W. The system was equipped with a multichannel plate and delay line detector, and the experiments were conducted under a vacuum level of 1 · 10$^{-9}$ mbar. High-resolution spectra were acquired with fixed analyzer pass energies of 20 eV. To prevent differential charging, the samples were positioned in the floating mode, and charge neutralization was applied to all samples as necessary. XANES spectra of H$_x$MoO$_3$ samples were collected using Si (311) crystal monochromators at the BL14W1 beamlines of the Shanghai Synchrotron Radiation Facility (SSRF) in Shanghai, China. Before the beamline analysis, the samples were compressed into thin sheets with a diameter of 1 cm and sealed using Kapton tape film. The X-ray Absorption Fine Structure (XAFS) spectra were acquired at room temperature utilizing a 4-channel Silicon Drift Detector (SDD), specifically the Bruker 5040 model. For Mo K-edge Extended X-ray Absorption Fine Structure (EXAFS) spectra, a transmission mode was employed. Minimal changes in line shape and peak position were observed in the Mo K-edge XANES spectra between successive scans conducted on a particular sample. Additionally, standard samples (Mo foil, MoO$_2$) were also subjected to transmission mode recording for their XAFS spectra. The obtained spectra were subjected to processing and analysis using the software tools Athena and Artemis.

### Electrochemical measurements

All the electrochemical measurements were conducted on a Biologic VMP-3 workstation at room temperature (20 °C). The standard three-electrode system Swagelok-type cell was used to evaluate the electrochemical performance of the electrodes, in which Ag/AgCl was used as the reference electrode, and active carbon (YP-50) film worked as the counter electrode. Phosphoric acid surfactant lyotropic liquid crystalline electrolyte (85% H$_3$PO$_4$ 4.33 grams and Polyoxyethylene (23) lauryl ether 1 gram were mixed in a sealed tube under 60 °C for 48 h) and 2 M H$_2$SO$_4$ solution worked as electrolytes for experiment and reference samples test. For the tests using organic acid, 1 M HOTf in ionic liquid (EMITFSI) worked as the electrolyte, and Ag/Ag$^+$ (filled with 0.01 M AgNO$_3$ and 0.1 M Tetrabutylammonium perchlorate in acetonitrile) worked as the reference electrode. The working electrodes of α-MoO$_3$ or CuFe-PBA were prepared by casting the slurry of micro-sized α-MoO$_3$ (70%) or CuFe-PBA (70%), conductive carbon Super C65 (20%), and PVDF (10%) on the carbon paper, and dried in the oven under 60 °C for 24 h. For the CS-AFM tests, the α-MoO$_3$ working electrodes were cast on the titanium foil with a ratio of 8:1:1. The mass loading of the working electrodes ranged from 1 to 3 mg cm$^{-2}$. All the electrodes and separator were punched into 12 mm discs. For the assembled full battery, the N/P ratio was 1.1. The Glass-fiber separator was immersed into the 60 °C PSL electrolyte and excluded air before assembling the cell.

### Computational methods

We performed density functional theory (DFT) calculations using the projector-augmented wave (PAW) method as implemented in the Vienna Ab initio Simulation Package. The generalized gradient approximation (GGA) with the Perdew–Burke–Ernzerhof (PBE) exchange-correlation functional was used. A uniform 4 × 4 × 2 k-mesh grid in the Brillouin zone was employed to optimize the crystal structures and calculate the projected density of states for MoO$_3$,

$HMoO_3$, $H_2MoO_3$, and $H_3MoO_3$. The plane-wave basis set cutoffs of the wave functions were set at 500 eV. The atomic positions of all crystal structures were fully relaxed until the forces on each atom are <0.01 eV/Å.

## Data availability

The data that support the findings of this study are presented in the paper and the Supplementary Information (PDF and Excel files). Source data are available. Source data are provided in this paper. Source data are provided with this paper.

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

## Acknowledgements

The research reported in this publication was supported by King Abdullah University of Science and Technology (KAUST). We thank Dr. Thom Leach, Amoeba Studios KAUST for making the illustrations.

## Author contributions

Y.J.L., W.L.Z., and J. Y. contributed equally to this work. H.N.A. supervised the project. H.N.A. and Y.J.L. conceived the idea. J. Y. performed the DFT calculations. Y.J.L. and W.L.Z. conducted the experiments and characterizations. Y.C.M., Z.M.Z., J.Y., Y.K., M.N.H., L.C., Y.Y.Y., and Q.X.W. conducted the sample characterizations and analysis. O.M.B., O.F.M., and X.X.Z. discussed the results. Y.J.L. drafted the manuscript and received comments from the other co-authors.

## Competing interests

The authors declare no competing interests.
