## [Peer Review File · Nature Communications]

Discovery of a Three-proton Insertion Mechanism in α -Molybdenum Trioxide Leading to Enhanced Charge Storage CapacityREVIEWER COMMENTS

Reviewer #1 (Remarks to the Author):

In this work, the authors report the proton-storage behavior of MoO₃ in a phosphoric acid surfactant lyotropic liquid crystalline electrolyte. A unique three-proton insertion mechanism is claimed, and one proton is indicated to be confined in the MoO₃ structure. In light of the metallic property and cracked structure of HMoO₃, the electrode depicts high-rate capability. The results are interesting to a certain degree. However, I cannot recommend its publication in Nat. Commun., as the potential impact is not strikingly attractive. I detailed my reason as follows.

1. Proton storage of MoO₃ has been widely reported in different aqueous electrolytes (10.1002/adma.202203335; 10.1021/jacs.2c03844; 10.1002/anie.202010073; 10.1002/anie.201803664). Compared with these early efforts, this study is more like an incremental effort as a specialized case. I did not detect any exciting advances in fundamental insights or in device technology.
2. The proton storage mechanism was not clearly disclosed and justified. Many open questions remain unclear in this report, such as what is the atomic structure and crystalline phase of the electrode after the first cycle? How do the Mo valence change and crystalline structure evolve during the three-plateau charge/discharge of MoO₃? What are the proton-storage sites in MoO₃ during the three-plateau process? All these questions should be answered with convincing evidence.
3. In Figure 3a, Mo 3d spectra look quite similar at -0.15 V, -0.45 V, and -0.65 V. The presence of Mo³⁺ is not justified by only XPS characterization.
4. 'During the first cycle, the pristine α -MoO₃ exhibits an interlayer distance of 6.92 Å, while it decreases to 6.75 Å in the protonation process and increases to 7.38 Å in the deprotonation process' It is not clear how the protonation process causes the interlayer distance decrease (6.75 Å), while partial protonation causes the obvious interlayer expansion (7.38 Å).
5. At low rates (1 A/g and 2 A/g in Figure 6b), the GCD curves show different curve shapes compared with curves at high rates. This observation implies that the anode/cathode mass balance is not well suited for low rates.
6. Long-term durability of the MoO₃ electrode in the three-electrode system should be tested at a low rate. Likely, the device life at a low rate should be evaluated.

Reviewer #2 (Remarks to the Author):

This article presents new and interesting results, where the authors demonstrated a record high capacity of MoO₃ in storing protons with a new lower-potential redox behavior revealed. As another interesting point, the proton insertion turns the compound from semiconducting to metallic. Overall, this article is of high quality. I recommend it be accepted after some minor revision.

The authors claim the three proton behavior; however, it does not seem that all three protons per MoO₃ can be reversibly extracted. From Figure 2c, it seems that only two protons are reversibly stored.

The authors compared the designed PSL electrolyte and 1 M H₃PO₄ electrolyte in Figure 1 b in terms of the voltage window. How about a concentrated H₃PO₄ with different concentrations?

What would be the “effective” concentration of H_3PO_4 in water when the surfactant is added?

The first protonation process is partially irreversible, where the lowest-potential plateau is not quite reversible. Do protons get trapped at low potentials? Or that is HER. Overall, the proton insertion/extraction coulombic efficiency is not very high for initial cycles. Why? And does the CE get improved over cycling?

For the storage mechanism of MoO_3 , it seems that the authors consider the storage of both “naked” protons as well as hydroniums, where the latter is used to explain the cracking behavior. Are all the inserted protons in the format of hydroniums? Or just partially.

From Figure 3f, it is not clear that the distance of Mo atoms is turned longer after proton insertion. Please point out the direction of the referred-to lattice.

Figure 6b, I don't think it is relevant to set the cutoff voltage of a battery to zero. The capacity below a voltage of 0.4 V is of very low quality, which should not be used to count the total capacity of a full cell.

The N/P ratio of the full cell should be given.

Response Letter to Reviewers

Manuscript Number: Nature Communications (NCOMMS-22-51746)

FULL TITLE: Discovery of a Three-proton Insertion Mechanism in α -Molybdenum Trioxide Leading to Significantly Enhanced Charge Storage Capacity

Reviewer Comments: Black

Authors' Responses: Blue

Revised Content: Red

REVIEWER COMMENTS

Reviewer #1 (Remarks to the Author):

In this work, the authors report the proton-storage behavior of MoO₃ in a phosphoric acid surfactant lyotropic liquid crystalline electrolyte. A unique three-proton insertion mechanism is claimed, and one proton is indicated to be confined in the MoO₃ structure. In light of the metallic property and cracked structure of HMoO₃, the electrode depicts high-rate capability. The results are interesting to a certain degree. However, I cannot recommend its publication in Nat. Commun., as the potential impact is not strikingly attractive. I detailed my reason as follows.

Authors' Reply:

We greatly appreciate your comments and valuable suggestions on the study regarding the proton storage of α -MoO₃. We have conducted additional work to provide

comprehensive responses and insights into these questions according to your valuable input.

Reviewer #1, Comment #1

1. Proton storage of MoO₃ has been widely reported in different aqueous electrolytes (10.1002/adma.202203335; 10.1021/jacs.2c03844; 10.1002/anie.202010073; 10.1002/anie.201803664). Compared with these early efforts, this study is more like an incremental effort as a specialized case. I did not detect any exciting advances in fundamental insights or in device technology.

Authors' Reply:

Thank you for your comments. The mentioned references have indeed made significant contributions to the proton storage field. However, we have uncovered a three-proton intercalation process and introduced groundbreaking advancements in fundamental insights when using α -MoO₃ as the anode material, which the researchers have not reported before. Thanks to the newly discovered redox phenomenon, MoO₃ exhibited a reversible high specific capacity of 362 mAh g⁻¹ as the anode in proton batteries (72% improvement in capacity and 230% improvement in specific energy density compared to the two-proton mechanism). Also, the fundamental science and energy storage mechanism behind the newly discovered three-proton intercalation process fills the gap in the existing literature, especially in proton chemistries, which may help to construct high-rate and high-energy aqueous energy storage devices in the future.

Actually, we have talked about these references in the introduction part of our manuscript. The achievements of these works as well as their limitations were listed below by publication date:

Reference 1: 10.1002/anie.201803664

In this work, Yan and co-workers have successfully showcased the electrochemical storage capability of hydrogen ions using molybdenum trioxide (α -MoO₃) as the anode. Their findings exhibit not only high coulombic efficiency but also exceptional stability, providing valuable insights into the potential application of α -MoO₃ as an effective storage medium for hydrogen ions. However, the proton-storage sites in α -MoO₃ and crystalline structure evolution during the charge and discharge process were not discussed.

Reference 2: 10.1002/anie.202010073

In this study, Lu and co-workers investigated the feasibility of manipulating electrochemical intercalating ions through interlayer engineering of α -MoO₃. The complete redox of Mo⁴⁺/Mo⁶⁺ of α -MoO₃ shows an attractive theoretical capacity of 372 mA h g⁻¹. These findings offer promising prospects for leveraging the unique properties of α -MoO₃ in energy storage applications.

However, the redox chemistry of α -MoO₃ in this study displayed a distinct behavior, as α -MoO₃ functioned as the cathode, setting it apart from prior research that employed α -MoO₃ as an anode in proton batteries. The resulting full battery (α -MoO₃/Zn) exhibited a lower energy density primarily due to the redox peaks of α -MoO₃ occurring in the low potential range. Additionally, this investigation focused only on the two-

proton intercalation process ($\text{Mo}^{4+}/\text{Mo}^{6+}$), which stands in contrast to the three-proton intercalation process observed in our studies.

Reference 3: 10.1002/adma.202203335

In this study, Yamada and co-workers successfully achieved the stabilization of α - MoO_3 as a cathode host material through the utilization of a super-concentrated $\text{Zn}^{2+}/\text{H}^+$ electrolyte. This approach enabled them to achieve a fully reversible (de)intercalation process involving 2.5 protons. These efforts highlight the potential of α - MoO_3 in energy storage applications and provide valuable insights into enhancing its cycling stability and proton intercalation capabilities. However, similar to the discussion presented in reference 2, this study also utilizes α - MoO_3 as the cathode, resulting in a low average discharge voltage and lower energy density attributed to the occurrence of redox peaks in the low potential range for α - MoO_3 . Furthermore, the α - MoO_3/Zn full battery necessitates an additional driving force to achieve reversible cycling, which is not conducive to practical applications.

Reference 4: 10.1021/jacs.2c03844

In this study, Jun and co-workers focus on investigating the potential of electrochemically activated metallic $\text{H}_{1.75}\text{MoO}_3$ nanobelts as electrodes for proton ion batteries. They leverage the advantages of Grotthuss proton conduction, which they have verified through ultralow activation energy measurements obtained from both experimental and theoretical analyses. However, the specific capacity of the $\text{H}_{1.75}\text{MoO}_3$ nanobelts was comparatively low compared to previous reports. Additionally,

H_{1.75}MoO₃ nanobelts exhibited significant dissolution in aqueous electrolytes, posing challenges to the long-term stability of the battery system.

More efforts have been done using α -MoO₃ as the anode to assemble the proton batteries that work in extreme environments, like 10.1038/s41467-022-33612-2, and 10.1021/acseenergylett.0c00109. But none of them talked about the three-proton intercalation process. We believe this newly discovered redox reaction of α -MoO₃ discussed in our manuscript offers fundamental insights into proton chemistries and will help to build high-rate and high-energy aqueous energy storage devices in the future.

Reviewer #1, Comment #2

2. The proton storage mechanism was not clearly disclosed and justified. Many open questions remain unclear in this report, such as what is the atomic structure and crystalline phase of the electrode after the first cycle? How do the Mo valence change and crystalline structure evolve during the three-plateau charge/discharge of MoO₃? What are the proton-storage sites in MoO₃ during the three-plateau process? All these questions should be answered with convincing evidence.

Authors' Reply:

Thank you for your comments. To comprehensively address the concerns raised by the reviewer, we have undertaken a substantial amount of additional work to provide comprehensive responses and insights into these questions.

Question 1 of comment 2: what is the atomic structure and crystalline phase of the electrode after the first cycle?

The atomic structure and crystalline phase of the MoO₃ electrodes after the initial cycle was investigated using a combination of low-dose high-resolution scanning transmission electron microscopy (HR-STEM), ex-situ X-ray diffraction (XRD) analysis, and density functional theory (DFT) calculations. In Fig. R1 below, lamella samples of MoO₃ in different charge states were prepared using focused ion beam (FIB) techniques for HR-STEM tests. Fig. R1a and b depict the HR-STEM images of pristine α -MoO₃, highlighting the distinct van der Waals gap (0.7 nm) observed between the zigzag arrays of molybdenum atoms, which aligns very well with the crystal structure (0.693 nm). Also, the selected area electron diffraction (SAED) patterns in Fig. R1a and b confirmed the orthorhombic symmetry of the pristine α -MoO₃.

In Fig. R1c and d, the observed interlayer space expansion (from 0.7 nm to 0.74 nm) and structural distortion (transformation from orthorhombic to monoclinic crystal structure) clearly indicate the effects of proton intercalation. The corresponding HR-STEM images and SAED patterns in Fig. R1c and d provide further evidence of the crystal structure distortion, specifically reflected in the change of the β angle from 90° to 93°. Notably, these experimental findings are in good agreement with the results obtained from DFT calculations, which yielded a similar β angle of 93.7°. Furthermore, the interlayer space expansion (0.693 nm to 0.742 nm) of α -MoO₃ after the first cycle was also confirmed by the XRD patterns, as shown in Fig. R2 below.

Fig. R1 Characterizations of the MoO_3 structure during protons (de)intercalation after the first cycle. **a** and **b** The crystal structure, HR-STEM, and SAED pattern (from left to right) of pristine $\alpha\text{-MoO}_3$ under different zoom axis (**a**: viewed along the [001] direction and **b**: viewed along the [010] direction). **c** and **d** The DFT simulated crystal structure, HR-STEM, and SAED pattern of HMoO_3 under different zoom axis (**c**: viewed along the [001] direction and **d**: viewed along the [010] direction).

Fig. R2 The ex-situ XRD curves of MoO₃ before and after the first cycle. **a** The pristine α -MoO₃ and **b** HMoO₃.

Question 2 of comment 2: How do the Mo valence change and crystalline structure evolve during the three-plateau charge/discharge of MoO₃?

The Mo valence changes of the MoO₃ electrodes during the discharge/charge process were investigated using a combination of ex-situ X-ray photoelectron spectroscopy (XPS) measurements and Mo K-edge X-ray absorption near-edge structure (XANES) spectra.

The Mo 3d spectrum (Fig. R3a, pristine) exhibited two pairs of 5/2–3/2 spin-orbit doublets. The intense doublet at 233.2 eV and 236.3 eV corresponded to the Mo⁶⁺ oxidation state, while the weaker doublet at 231.2 eV and 234.3 eV corresponded to the Mo⁵⁺ oxidation state. Upon discharging the MoO₃ electrode to 0.35 V after the initial cycle, the presence of Mo⁶⁺ and Mo⁵⁺ oxidation states, along with a third oxidation state (Mo⁴⁺), was observed. This result confirmed the trapping of protons in HMoO₃. At lower potentials, further reduction of Mo⁶⁺ and Mo⁵⁺ occurred, leading to an increase in the proportion of the Mo⁴⁺ state, indicating the intercalation of more protons (Fig. R3a, -0.15 V: sample H_{1.86}MoO₃).

Fig. R3 Characterizations of the Mo valence states in MoO_3 electrodes during protons (de)intercalation process. **a** The Mo 3d XPS spectrums of MoO_3 electrodes performed at different discharge states. **b** The XANES spectra of H_xMoO_3 compared with Mo foil. **c** The XANES spectra of $\text{H}_{2.15}\text{MoO}_3$ and $\text{H}_{2.83}\text{MoO}_3$ compared with Mo foil and MoO_2 reference.

However, in the discharge state at -0.45 V (Sample $\text{H}_{2.15}\text{MoO}_3$), two new peaks appeared in the spectrum at 229.3 eV and 232.6 eV, which could be assigned to the Mo^{3+} oxidation state. Upon further reduction of MoO_3 to -0.65 V (Sample $\text{H}_{2.83}\text{MoO}_3$), a higher content of Mo^{3+} was observed, in line with the expected third proton insertion. These findings provide evidence of the evolution of oxidation states (Mo^{6+} , Mo^{5+} , Mo^{4+} , and Mo^{3+}) in the MoO_3 electrode during the discharge process, supporting the progressive intercalation of protons.

Considering the limited depth sensitivity of the XPS test on the nanometer scale and the sample size being in the order of tens of microns, further analysis of the valence

state of the Mo element in the sample was conducted using XANES spectroscopy. The Mo K-edge XANES spectra clearly indicated a shift towards lower energy absorption near the edge of $H_x\text{MoO}_3$ upon proton intercalation, compared to pristine $\alpha\text{-MoO}_3$, signifying a reduced oxidation state of Mo in the MoO_3 electrodes (Fig. R3b).

To further validate the presence of the Mo^{3+} state in the $\text{H}_{2.15}\text{MoO}_3$ and $\text{H}_{2.83}\text{MoO}_3$ samples, we introduced a MoO_2 reference for comparison (Fig. R3c). The absorption near the edge of $\text{H}_{2.15}\text{MoO}_3$ exhibited a slight shift towards the lower energy region, with a portion of the edge curve remaining at higher energy compared to the MoO_2 reference. This observation confirmed the reduced oxidation state of Mo in the $\text{H}_{2.15}\text{MoO}_3$ electrode, consistent with the presence of the Mo^{3+} state. In contrast, the absorption near the edge of $\text{H}_{2.83}\text{MoO}_3$ displayed a significant shift towards the lower energy region, clearly distinguishing it from both the $\text{H}_{2.15}\text{MoO}_3$ electrode and the MoO_2 reference. This notable shift confirmed a substantial abundance of the Mo^{3+} state in $\text{H}_{2.83}\text{MoO}_3$, providing further support for the phenomenon of the third proton insertion.

Overall, the XANES spectroscopy results not only addressed the limitations of XPS but also provided compelling evidence for the reduced oxidation state of Mo and the presence of the Mo^{3+} state in the MoO_3 electrodes upon proton intercalation.

The evolution of atomic structure and crystalline phase of the MoO_3 electrodes upon the discharge/charge process was investigated using a combination of low-dose HR-STEM, ex-situ XRD analysis, and DFT calculations. To perform HR-STEM tests, lamella samples of $\text{H}_{1.86}\text{MoO}_3$, $\text{H}_{2.15}\text{MoO}_3$, and $\text{H}_{2.83}\text{MoO}_3$ were prepared using the FIB

technique, as depicted in Fig. R4 to R6. Fig. R4b and R4e illustrate the reduced interlayer spacing in $H_{1.86}MoO_3$ (from 0.74 nm to 0.71 nm compared to $HMoO_3$) and clearly demonstrate the structural distortion resulting from proton intercalation. The corresponding SAED pattern in Fig. R4f provides additional supporting evidence of the crystal structure distortion, manifested by the change in the β angle from 93° to 93.5° . Remarkably, these experimental observations are consistent with the results obtained from DFT calculations, which yielded a similar β angle of 93.8° .

Fig. R4 Characterizations of the MoO_3 structure during protons (de)intercalation process. **a to c** The DFT simulated crystal structure, HR-STEM, and SAED pattern of $H_{1.86}MoO_3$ viewed along the $[001]$ direction. **d to f** The DFT simulated crystal structure, HR-STEM, and SAED pattern of $H_{1.86}MoO_3$ viewed along the $[010]$ direction.

Similarly, the HR-STEM investigations conducted on $H_{2.15}MoO_3$ and $H_{2.83}MoO_3$ (as shown in Figures R5 and R6) revealed a further reduction in the interlayer spacing (From 0.71 nm to 0.68 nm and 0.69 nm, respectively) and pronounced structural distortion, thus reinforcing the significant influence of proton intercalation.

Additionally, the corresponding SAED patterns in Fig. R5f and R6f offer supplementary evidence of the crystal structure distortion, notably reflected in the changes of the β angle from 93.5° to 94° and subsequently to 94.5° .

Fig. R5 Characterizations of the MoO_3 structure during protons (de)intercalation process. **a to c** The DFT simulated crystal structure, HR-STEM and SAED pattern of $\text{H}_{2.15}\text{MoO}_3$ viewed along the $[001]$ direction. **d to f** The DFT simulated crystal structure, HR-STEM and SAED pattern of $\text{H}_{2.15}\text{MoO}_3$ viewed along the $[010]$ direction.

Moreover, the ex-situ XRD pattern refinement provided valuable structural information and refined lattice parameters of H_xMoO_3 , as illustrated in Fig R2 and R7 to R10. These findings have been compiled in Table R1, demonstrating a close agreement with the results obtained from the HR-STEM investigations.

Overall, the comprehensive analysis involving HR-STEM, XRD analysis, and DFT calculations provides compelling insights into the evolution of the atomic structure and crystalline phase of the MoO_3 electrodes during the discharge/charge process.

Fig. R6 Characterizations of the MoO_3 structure during protons (de)intercalation process. **a to c** The DFT simulated crystal structure, HR-STEM, and SAED pattern of $\text{H}_{2.83}\text{MoO}_3$ viewed along the [001] direction. **d to f** The DFT simulated crystal structure, HR-STEM, and SAED pattern of $\text{H}_{2.83}\text{MoO}_3$ viewed along the [010] direction.

Fig. R7 Cyclic voltammety curves of MoO_3 electrodes in a three-electrode system. **a** The first cycle and **b** the second cycle. The marked point indicated the cut-off potential for ex-situ XRD tests.

Fig. R8 The ex-situ XRD curves of MoO_3 during protons (de)intercalation process. **a** The $H_{1.86}MoO_3$ and **b** $H_{2.15}MoO_3$.

Fig. R9 The ex-situ XRD curves of MoO_3 during protons (de)intercalation process. **a** The $H_{2.83}MoO_3$ and **b** $H_{1.68}MoO_3$.

Fig. R10 The ex-situ XRD curves of MoO_3 during protons (de)intercalation process ($H_{2.1}MoO_3$).

Tab. R1 The structural information and refined lattice parameters of $H_x\text{MoO}_3$ obtained from ex-situ XRD pattern refinement.

Stage	Phase	Space group	a/Å	b/Å	c/Å	Vol/Å ³	Beta/°
i	$\alpha\text{-MoO}_3$	Pnma	13.8638	3.7004	3.9649	203.41	
ii	$H_{1.68}\text{MoO}_3$	C2/m	13.9312	3.7752	4.0717	213.756	94.48
iii	$H_{2.83}\text{MoO}_3$	C	13.7179	3.8614	4.0661	214.5206	95.14
iv	$H_{2.1}\text{MoO}_3$	C	13.6057	3.8674	4.058	212.6369	95.36
v	HMoO_3	I2/m	14.0808	3.7163	7.4735	390.1492	93.96
vi	$H_{1.86}\text{MoO}_3$	C2/m	13.9811	3.7730	4.0516	213.2424	93.85
vii	$H_{2.15}\text{MoO}_3$	C	13.5835	3.8979	4.0382	213.0686	94.78

Question 3 of comment 2: What are the proton-storage sites in MoO_3 during the three-plateau process?

The pristine $\alpha\text{-MoO}_3$ involves three distinct types of oxygen: the terminating oxygen (O_1) coordinated to a single Mo atom, the bridging oxygen (O_2) coordinated to two Mo atoms, and the oxygen shared among three Mo atoms (O_3), as shown in Fig. R11. Based on the DFT calculations shown in Fig. R12, it is revealed that among the oxygen atoms present in the Fig. R11, O_1 and O_2 exhibit potential as proton storage sites, whereas O_3 is unsuitable for proton coordination due to its coordination with three molybdenum atoms. The intercalation sequence of protons in MoO_3 is as follows: in HMoO_3 , the most stable state is achieved when protons occupy site 1. Similarly, in H_2MoO_3 , the most stable state occurs when protons reside at sites 1-1 and 1-2, while sites 2-1 and 2-2 are not coordinated by protons. Lastly, in H_3MoO_3 , the third proton has same probability of inserting into site 2-1 or 2-2.

Fig. R11 The illustration crystal structural arrangement of oxygen atoms in α - MoO_3 , highlighting three distinct types: the terminating oxygen (O_1) coordinated to a single Mo atom, the bridging oxygen (O_2) coordinated to two Mo atoms, and the oxygen shared among three Mo atoms (O_3).

Fig. R12 Illustration of the DFT calculation results, highlighting the probable locations of proton storage sites in H_xMoO_3 .

To validate the DFT simulation results, an extended X-ray absorption fine structure (EXAFS) analysis was conducted to gain further insights into the Mo-O coordination, and the results are summarized in Table R3. According to the theoretical Mo-O distances listed in Table R2, the Fourier transform EXAFS (FT-EXAFS) curves of

$H_x\text{MoO}_3$ exhibit two prominent peaks corresponding to the nearest Mo-O₁ and the next nearest Mo-O₂ (1.738 Å) coordination, respectively (Fig. R14 to R18).

After the first cycle of the $\alpha\text{-MoO}_3$ electrode, the Mo-O₁ distance in HMoO₃ significantly decreases, while the Mo-O₂ distance in HMoO₃ slightly decreases (Fig. R15a) compared to the Mo-O distances in pristine $\alpha\text{-MoO}_3$ (Fig. R14a). These findings confirm the occupation of site 1 by protons in the HMoO₃ composite, which aligns with the DFT calculation results. The decreased Mo-O distances also indicate the trapping of protons in HMoO₃ after the first cycle, signifying a more stable state than $\alpha\text{-MoO}_3$, which represents an irreversible proton intercalation process.

Subsequent proton intercalation in $H_{1.86}\text{MoO}_3$ (Fig. R16 a) results in a slight decrease in the Mo-O₁ distance, while the Mo-O₂ distance remains the same compared to HMoO₃ (Fig. R15 a). These observations confirm that protons occupy sites 1-1 and 1-2 in the $H_{1.86}\text{MoO}_3$ composite. However, upon further proton intercalation in $H_{2.15}\text{MoO}_3$, the intense peak corresponding to Mo-O₁ disappears, and the second main peak (Mo-O₂) shifts to a lower distance. This behavior is attributed to the terminal oxygen (O₁) being likely to exist in the state of free water after combining with two protons, resulting in weak coordination with the Mo atom, ultimately leading to its disappearance. Additionally, the decreased Mo-O₂ distance confirms the storage of the third proton on site 2. Further protons intercalation in $H_{2.83}\text{MoO}_3$ (Fig. R18 a) leads to a slight decrease in the Mo-O₂ distance compared to $H_{2.15}\text{MoO}_3$, confirming the storage of the third proton on the bridging oxygen (O₂).

The combination of DFT calculations and EXAFS analysis provides valuable insights into the proton storage sites within MoO₃ electrodes during the discharge/charge process.

According to these additional work and the suggestions from the reviewer, we have revised the manuscript and added these figure into Supporting document as following:

The Mo 3d spectrum (Fig. R3a, pristine) exhibited two pairs of 5/2–3/2 spin-orbit doublets. The intense doublet at 233.2 eV and 236.3 eV corresponded to the Mo⁶⁺ oxidation state, while the weaker doublet at 231.2 eV and 234.3 eV corresponded to the Mo⁵⁺ oxidation state.³⁹ Upon discharging the MoO₃ electrode to 0.35 V after the initial cycle, the presence of Mo⁶⁺ and Mo⁵⁺ oxidation states, along with a third oxidation state (Mo⁴⁺), was observed. This result confirmed the trapping of protons in HMoO₃. At lower potentials, further reduction of Mo⁶⁺ and Mo⁵⁺ occurred, leading to an increase in the proportion of the Mo⁴⁺ state, indicating the intercalation of more protons (Fig. R3a, -0.15 V: sample H_{1.86}MoO₃). However, in the discharge state at -0.45 V (Sample H_{2.15}MoO₃), two new peaks appeared in the spectrum at 229.3 eV and 232.6 eV, which could be assigned to the Mo³⁺ oxidation state. Upon further reduction of MoO₃ to -0.65 V (Sample H_{2.83}MoO₃), a higher content of Mo³⁺ was observed, in line with the expected third proton insertion. These findings provide evidence of the evolution of oxidation states (Mo⁶⁺, Mo⁵⁺, Mo⁴⁺, and Mo³⁺) in the MoO₃ electrode during the discharge process, supporting the progressive intercalation of protons.

Considering the limited depth sensitivity of the XPS test on the nanometer scale and the sample size being in the order of tens of microns, further analysis of the valence

state of the Mo element in the $H_x\text{MoO}_3$ samples was conducted using Mo K-edge X-ray absorption near-edge structure (XANES) spectroscopy. The XANES spectra clearly indicated a shift towards lower energy absorption near the edge of $H_x\text{MoO}_3$ upon proton intercalation, compared to pristine $\alpha\text{-MoO}_3$, signifying a reduced oxidation state of Mo in the MoO_3 electrodes (Fig. 3d). To further validate the presence of the Mo^{3+} state in the $\text{H}_{2.15}\text{MoO}_3$ and $\text{H}_{2.83}\text{MoO}_3$ samples, we introduced a MoO_2 reference for comparison (Supplementary Fig. 12). The absorption near the edge of $\text{H}_{2.15}\text{MoO}_3$ exhibited a slight shift towards the lower energy region, with a portion of the edge curve remaining at higher energy compared to the MoO_2 reference. This observation confirmed the reduced oxidation state of Mo in the $\text{H}_{2.15}\text{MoO}_3$ electrode, consistent with the presence of the Mo^{3+} state. In contrast, the absorption near the edge of $\text{H}_{2.83}\text{MoO}_3$ displayed a significant shift towards the lower energy region, clearly distinguishing it from both the $\text{H}_{2.15}\text{MoO}_3$ electrode and the MoO_2 reference. This notable shift confirmed a substantial abundance of the Mo^{3+} state in $\text{H}_{2.83}\text{MoO}_3$, providing further support for the phenomenon of the third proton insertion. Overall, the XANES spectroscopy results not only addressed the limitations of XPS but also provided compelling evidence for the reduced oxidation state of Mo and the presence of the Mo^{3+} state in the MoO_3 electrodes upon proton intercalation.

The evolution of atomic structure and crystalline phase of the MoO_3 electrodes upon the discharge/charge process was investigated using a combination of low-dose high-resolution scanning transmission electron microscopy (HR-STEM), X-ray diffraction (XRD) analysis, and density functional theory (DFT) calculations. The in-

situ XRD patterns (Fig. 3b) indicated that the interlayer spacing of MoO₃ showed a periodic change with high stability, which was consistent with the reversible charge-discharge curve after the first cycle. During the first cycle, the pristine α -MoO₃ exhibits an interlayer distance of 6.92 Å, while it decreases to 6.75 Å in the protonation process and increases to 7.38 Å in the deprotonation process (Fig. 3c and Supplementary Fig. 13). This irreversible change in lattice spacing is attributed to the fact that some of the protons were not extracted during deprotonation, which was also observed in literature work using H₂SO₄ electrolyte.³⁵

The lamella samples of H_xMoO₃ in different charge states were prepared using focused ion beam (FIB) techniques for HR-STEM tests. Fig. 4a and b depict the HR-STEM images of pristine α -MoO₃, highlighting the distinct van der Waals gap (0.7 nm) observed between the zigzag arrays of molybdenum atoms, which aligns very well with the crystal structure (0.693 nm). Also, the selected area electron diffraction (SAED) patterns in Fig. 4a and b confirmed the orthorhombic symmetry of the pristine α -MoO₃. In Fig. 4c and d, the observed interlayer space expansion (from 0.7 nm to 0.74 nm) and structural distortion (transformation from orthorhombic to monoclinic crystal structure) clearly indicate the effects of proton intercalation.³⁵ The corresponding HR-STEM images and SAED patterns in Fig. 4c and d provide further evidence of the crystal structure distortion, specifically reflected in the change of the β angle from 90° to 93°. Notably, these experimental findings are in good agreement with the results obtained from DFT calculations, which yielded a similar β angle of 93.7°. Furthermore, lamella samples of H_{1.86}MoO₃, H_{2.15}MoO₃, and H_{2.83}MoO₃ were prepared using the FIB

technique, as depicted in Supplementary Fig. 14 to 16. Supplementary Fig. 14b and e illustrates the reduced interlayer spacing in $\text{H}_{1.86}\text{MoO}_3$ (from 0.74 nm to 0.71 nm compared to HMoO_3) and clearly demonstrate the structural distortion resulting from proton intercalation. The corresponding SAED pattern in Supplementary Fig. 14f provides additional supporting evidence of the crystal structure distortion, manifested by the change in the β angle from 93° to 93.5° . Remarkably, these experimental observations are consistent with the results obtained from DFT calculations, which yielded a similar β angle of 93.8° . Similarly, the HR-STEM investigations conducted on $\text{H}_{2.15}\text{MoO}_3$ and $\text{H}_{2.83}\text{MoO}_3$ (as shown in Supplementary Fig. 15 and 16) revealed a further reduction in the interlayer spacing (From 0.71 nm to 0.68 nm and 0.69 nm, respectively) and pronounced structural distortion, thus reinforcing the significant influence of proton intercalation. Additionally, the corresponding SAED patterns in Supplementary Fig. 15f and 16f offer supplementary evidence of the crystal structure distortion, notably reflected in the changes of the β angle from 93.5° to 94° and subsequently to 94.5° . Moreover, the ex-situ XRD pattern refinement provided valuable structural information and refined lattice parameters of H_xMoO_3 , as illustrated in Supplementary Fig. 17 to 21. These findings have been compiled in Supplementary Table 1, demonstrating a close agreement with the results obtained from the HR-STEM investigations. Overall, the comprehensive analysis involving HR-STEM, XRD analysis, and DFT calculations provides compelling insights into the evolution of the atomic structure and crystalline phase of the MoO_3 electrodes during the discharge/charge process.

Moreover, the proton-storage sites in $H_x\text{MoO}_3$ during the discharge/charge process was investigated using the DFT simulation and extended X-ray absorption fine structure (EXAFS) analysis. Based on the DFT calculations shown in Supplementary Fig. 22, it is revealed that O_1 and O_2 exhibit potential as proton storage sites, whereas O_3 is unsuitable for proton coordination due to its coordination with three molybdenum atoms. The intercalation sequence of protons in MoO_3 is as follows: in HMoO_3 , the most stable state is achieved when protons occupy site 1. Similarly, in H_2MoO_3 , the most stable state occurs when protons reside at sites 1-1 and 1-2, while sites 2-1 and 2-2 are not coordinated by protons. Lastly, in H_3MoO_3 , the third proton has same probability of inserting into site 2-1 or 2-2.

To validate the DFT simulation results, the EXAFS analysis was conducted to gain further insights into the Mo-O coordination (Supplementary Fig. 23 to 28), and the results are summarized in Supplementary Table 2. According to the theoretical Mo-O distances listed in Supplementary Table 3, the Fourier transform EXAFS (FT-EXAFS) curves of $H_x\text{MoO}_3$ exhibit two prominent peaks corresponding to the nearest Mo- O_1 and the next nearest Mo- O_2 (1.738 Å) coordination, respectively. After the first cycle of the $\alpha\text{-MoO}_3$ electrode, the Mo- O_1 distance in HMoO_3 significantly decreases, while the Mo- O_2 distance in HMoO_3 slightly decreases (Supplementary Fig. 25) compared to the Mo-O distances in pristine $\alpha\text{-MoO}_3$ (Supplementary Fig. 24). These findings confirm the occupation of site 1 by protons in the HMoO_3 composite, which aligns with the DFT calculation results. The decreased Mo-O distances also indicate the trapping

of protons in HMoO_3 after the first cycle, signifying a more stable state than $\alpha\text{-MoO}_3$, which represents an irreversible proton intercalation process.

Subsequent proton intercalation in $\text{H}_{1.86}\text{MoO}_3$ (Supplementary Fig. 26) results in a slight decrease in the Mo-O₁ distance, while the Mo-O₂ distance remains the same compared to HMoO_3 . These observations confirm that protons occupy sites 1-1 and 1-2 in the $\text{H}_{1.86}\text{MoO}_3$ composite. However, upon further proton intercalation in $\text{H}_{2.15}\text{MoO}_3$ (Supplementary Fig. 27), the intense peak corresponding to Mo-O₁ disappears, and the second main peak (Mo-O₂) shifts to a lower distance. This behavior is attributed to the terminal oxygen (O₁) being likely to exist in the state of water after combining with two protons, resulting in weak coordination with the Mo atom, ultimately leading to its disappearance. Additionally, the decreased Mo-O₂ distance confirms the storage of the third proton on site 2. Further protons intercalation in $\text{H}_{2.83}\text{MoO}_3$ (Supplementary Fig. 28) leads to a slight decrease in the Mo-O₂ distance compared to $\text{H}_{2.15}\text{MoO}_3$, confirming the storage of the third proton on the bridging oxygen (O₂). Overall, the combination of DFT calculations and EXAFS analysis provides valuable insights into the proton storage sites within MoO_3 electrodes during the discharge/charge process.

Fig. 3 Characterization of the α - MoO_3 structure during proton (de)intercalation. a The Mo 3d XPS spectrums of H_xMoO_3 electrodes performed at different discharge states. b The in-situ XRD curves upon protons insertion and extraction in PSL electrolyte (82 minutes per cycle). c The interlayer spacing of MoO_3 derived from the in-situ XRD curves. d The XANES spectra of H_xMoO_3 compared with Mo foil. e EXAFS spectra and fitting curve of H_xMoO_3 in R space.

Tab. R2 The theoretical Mo-O distances of α - MoO_3 .

Mo-O	Distances (\AA)
Mo-O ₁	1.678
Mo-O ₂ ^I	1.738
Mo-O ₂ ^{II}	2.244
Mo-O ₃ ^I	1.949
Mo-O ₃ ^{II}	1.949
Mo-O ₃ ^{III}	2.327

Fig. R13 **a** The FT-EXAFS spectra with the fitting result. **b** EXAFS spectra and fitting curve of Mo K-edge for Mo foil in R space. **c** Wavelet transform for the k^2 -weighted EXAFS signal of Mo foil.

Fig. R14 **a** The FT-EXAFS spectra with the fitting result. **b** EXAFS spectra and fitting curve of Mo K-edge for α -MoO₃ in R space. **c** Wavelet transform for the k^2 -weighted EXAFS signal of α -MoO₃.

Fig. R15 **a** The FT-EXAFS spectra with the fitting result. **b** EXAFS spectra and fitting curve of Mo K-edge for HMoO₃ in R space. **c** Wavelet transform for the k^2 -weighted EXAFS signal of HMoO₃.

Fig. R16 **a** The Fourier transform EXAFS spectra with the fitting result. **b** EXAFS spectra and fitting curve of Mo K-edge for $H_{1.86}MoO_3$ in R space. **c** Wavelet transform for the k^2 -weighted EXAFS signal of $H_{1.86}MoO_3$.

Fig. R17 **a** The Fourier transform EXAFS spectra with the fitting result. **b** EXAFS spectra and fitting curve of Mo K-edge for $H_{2.15}MoO_3$ in R space. **c** Wavelet transform for the k^2 -weighted EXAFS signal of $H_{2.15}MoO_3$.

Fig. R18 **a** The Fourier transform EXAFS spectra with the fitting result. **b** EXAFS spectra and fitting curve of Mo K-edge for $H_{2.83}MoO_3$ in R space. **c** Wavelet transform for the k^2 -weighted EXAFS signal of $H_{2.83}MoO_3$.

Table. R3 Structural parameters of different samples extracted from the EXAFS fitting.

Sample	Shell	CN	R (Å)	σ^2 (10^{-3}Å^2)	ΔE_0 (eV)	R factor
Mo foil	Mo-Mo	12 (fixed)	2.85±0.01	1.7±1.6	6.1±1.4	0.009
α -MoO ₃	Mo-O	5.6±0.7	2.18±0.01	1.1±1.2	4.9±3.2	0.014
HMoO ₃	Mo-O	4.9±1.3	2.16±0.01	11.1±7.9	4.9±1.9	0.009
H _{1.86} MoO ₃	Mo-O	4.4±0.2	2.10±0.00	1.1±1.2	2.7±8.1	0.020
H _{2.15} MoO ₃	Mo-O	3.9±0.5	2.02±0.01	12.7±2.9	1.3±1.9	0.014
H _{2.83} MoO ₃	Mo-O	3.2±0.3	2.01±0.01	2.3±8.1	1.5±2.5	0.013

Reviewer #1, Comment #3

3. In Figure 3a, Mo 3d spectra look quite similar at -0.15 V, -0.45 V, and -0.65 V. The presence of Mo³⁺ is not justified by only XPS characterization.

Thank you for your comments. To comprehensively address the concerns raised by the reviewer, we have conducted additional work to provide comprehensive responses and insights into these questions.

To further validate the presence of the Mo³⁺ state in the H_{2.15}MoO₃ and H_{2.83}MoO₃ samples, analysis of the valence state of the Mo element in the sample was conducted using XANES spectroscopy. To ensure the reliability of the results, a MoO₂ reference was introduced for comparison, as shown in Fig. R3c. The absorption near the edge of H_{2.15}MoO₃ exhibited a slight shift towards the lower energy region, with a portion of the edge curve remaining at higher energy compared to the MoO₂ reference. This observation confirmed the reduced oxidation state of Mo in the H_{2.15}MoO₃ electrode, consistent with the presence of the Mo³⁺ state. In contrast, the absorption near the edge of H_{2.83}MoO₃ displayed a significant shift towards the lower energy region, distinguishing it from the H_{2.15}MoO₃ electrode and the MoO₂ reference. This notable

shift confirmed a substantial abundance of the Mo^{3+} state in $\text{H}_{2.83}\text{MoO}_3$, providing further support for the phenomenon of the third proton insertion.

Thus, the XANES spectroscopy results effectively addressed the limitations of XPS and provided compelling evidence supporting the reduced oxidation state of Mo and the presence of the Mo^{3+} state in the MoO_3 electrodes during the proton intercalation.

Reviewer #1, Comment #4

4. ‘During the first cycle, the pristine $\alpha\text{-MoO}_3$ exhibits an interlayer distance of 6.92 Å, while it decreases to 6.75 Å in the protonation process and increases to 7.38 Å in the deprotonation process’ It is not clear how the protonation process causes the interlayer distance decrease (6.75 Å), while partial deprotonation causes the obvious interlayer expansion (7.38 Å).

Thank you for your comments. To comprehensively address the concerns raised by the reviewer, we have conducted additional work and did some research to provide comprehensive responses and insights into these questions.

The HR-STEM investigation performed on $\alpha\text{-MoO}_3$ and $\text{H}_{2.15}\text{MoO}_3$ demonstrated a decrease in interlayer spacing (from 0.7 nm to 0.68 nm) along with significant structural distortion, as depicted in Fig. R1 and Fig. R5. The reduction in the interlayer distance was further validated through ex-situ XRD tests, as indicated in Table R1. After protons intercalation, the presence of hydrogen bonds enhances the interlayer van der Waals interaction, leading to a decrease in the interlayer distance.¹ While during the deprotonation process, the increased interlayer space (from 0.68 nm to 0.74 nm) was

confirmed by the HR-STEM tests (Fig. R1, HMoO_3), and the increased atomic distance probably results from the breaking of hydrogen bonds during deprotonation process, which weakened the interlayer van der Waals interaction. Furthermore, the pronounced structural distortion observed in the Mo layers played a crucial role in the expansion of the interlayer space. This distortion caused a reduction in the distance between neighboring zigzag arrays of molybdenum atoms due to the structural perturbation, resulting in an enlargement of the van der Waals gap.¹

Reviewer #1, Comment #5

5. At low rates (1 A/g and 2 A/g in Figure 6b), the GCD curves show different curve shapes compared with curves at high rates. This observation implies that the anode/cathode mass balance is not well suited for low rates.

Thank you for your comments. To comprehensively address the concerns raised by the reviewer, we have conducted additional work to provide comprehensive responses and insights into these questions.

To investigate the potential impact of anode/cathode mass balance on the galvanostatic charge-discharge (GCD) curves, a full battery with a N/P ratio of 1.5 (compared to a N/P ratio of 1.1 in Fig. 6b) was assembled. Remarkably, similar results were obtained when compared to the previous data (Fig. R20 a), suggesting that the shape of the GCD curves remained unaffected by the anode/cathode mass balance. Further investigations revealed that the CuFe-PBA cathode electrode exhibited a rapid capacity decline in the low potential region as the current density increased (Fig. R20 b). This behavior can be attributed to the sluggish kinetics of the $\text{Cu}^+/\text{Cu}^{2+}$ redox

reaction (Fig. R20 c), hindering its effectiveness at high current densities and resulting in the reduction of electrode capacity.

Fig. R19 **a** GCD curves of the full battery recorded at different current densities (with N/P ratio of 1.5). **b** GCD curves of the CuFe-PBA cathode recorded at different current densities. **c** CV curves of CuFe-PBA cathode electrode in a three-electrode system.

Reviewer #1, Comment #6

6. Long-term durability of the MoO₃ electrode in the three-electrode system should be tested at a low rate. Likely, the device life at a low rate should be evaluated.

Thank you for your comments. To comprehensively address the comments raised by the reviewer, we have conducted additional work to provide comprehensive responses and insights into these questions.

The cycling performance of the MoO₃ electrode was evaluated at a low current density using the specially designed PSL electrolyte. The electrode exhibited a capacity retention of 77% after 500 cycles (Fig. R20 a). Notably, a significant decrease in specific discharge capacity was observed after 300 cycles. This reduction can be attributed to the formation of hydrogen cavities on the electrode surface due to the hydrogen evolution reaction (HER). The presence of these cavities hinders the diffusion of protons at the electrode interface, resulting in a sharp decline in capacity. It should be noted that complete suppression of the HER is challenging, particularly during low

current discharge, where the HER on the current collector and conductive carbon remains active (Fig. R21). Therefore, optimization of the electrolyte design, such as utilizing a solid proton electrolyte, is essential to achieve a stable charge-discharge cycle at low current density.² Similar findings were obtained when cycling the full battery at low current density, with a capacity retention of 78% after 500 cycles (Fig R20 b). Moving forward, we will continue to focus on optimizing the proton electrolyte to achieve stable cycling of the proton battery under low current conditions.

Fig. R20 **a** Cycling performance of MoO₃ electrode at 1 A g⁻¹ using the designed PSL electrolyte. **b** Cycling performance of full battery at 0.5 A g⁻¹ using the designed PSL electrolyte.

Fig. R21 CV curve of MoO₃ electrode at 1 mV s⁻¹ using the designed PSL electrolyte.

Reference list

- 1 Xu, W. W. et al. Proton Storage in Metallic $\text{H}_{1.75}\text{MoO}_3$ Nanobelts through the Grotthuss Mechanism. *J. Am. Chem. Soc.* **144**, 17407-17415 (2022).
- 2 Wang, S. et al. Acid-in-Clay Electrolyte for Wide-Temperature-Range and Long-Cycle Proton Batteries. *Adv Mater.* **23**, 2202063 (2022).

Reviewer #2 (Remarks to the Author):

This article presents new and interesting results, where the authors demonstrated a record high capacity of MoO₃ in storing protons with a new lower-potential redox behavior revealed. As another interesting point, the proton insertion turns the compound from semiconducting to metallic. Overall, this article is of high quality. I recommend it be accepted after some minor revision.

Authors' Reply:

We greatly appreciate your comments and valuable suggestions on the study regarding the proton storage of α -MoO₃. We have conducted additional work to provide comprehensive responses and insights into these questions according to your valuable input.

Reviewer #2, Comment #1

1. The authors claim the three-proton behavior; however, it does not seem that all three protons per MoO₃ can be reversibly extracted. From Figure 2c, it seems that only two protons are reversibly stored.

Thank you for your comments. To comprehensively address the comments raised by the reviewer, we have conducted additional work to provide comprehensive responses and insights into these questions.

It is true that only two protons are reversibly stored during the discharge/charge process, and the three-proton behavior only occurred during the first discharge process, 538 mAh g⁻¹ specific capacity was observed during this first discharge process, which

was equivalent to 2.89 mol H⁺ inserted per MoO₃ formula unit, very close to the theoretical maximum capacity of 558.6 mAh g⁻¹ (3 mol H⁺ per MoO₃ formula unit). To validate the proton trapped nature in the HMO₃, a combination of ex-situ X-ray photoelectron spectroscopy (XPS) measurements and Mo K-edge X-ray absorption near-edge structure (XANES) spectra analysis were conducted.

The Mo 3d spectrum (Fig. R1 a) exhibited two pairs of 5/2–3/2 spin-orbit doublets. The intense doublet at 233.2 eV and 236.3 eV corresponded to the Mo⁶⁺ oxidation state, while the weaker doublet at 231.2 eV and 234.3 eV corresponded to the Mo⁵⁺ oxidation state. Upon discharging the MoO₃ electrode to 0.35 V after the initial cycle, the presence of Mo⁶⁺ and Mo⁵⁺ oxidation states, along with a third oxidation state (Mo⁴⁺), was observed (Fig. R1 a). This result confirmed the trapping of protons in HMO₃.

Considering the limited depth sensitivity of the XPS test on the nanometer scale and the sample size being in the order of tens of microns, further analysis of the valence state of the Mo element in the sample was conducted using XANES spectroscopy. The Mo K-edge XANES spectra clearly indicated a shift towards lower energy absorption near the edge of H_xMoO₃ upon proton intercalation, compared to pristine α-MoO₃, signifying a reduced oxidation state of Mo in the MoO₃ electrodes (Fig. R1 b). To further validate the trapped protons in the HMO₃ sample, we introduced the pristine α-MoO₃ and Mo foil references for comparison (Fig. R1 c). The absorption near the edge of HMO₃ exhibited a significant shift towards the lower energy region, compared to the pristine α-MoO₃. This observation confirmed the trapping of protons in HMO₃.

According to this additional work and the concerns from the reviewer, we have revised the manuscript and added these figure R1 into manuscript and the Supplementary information.

Considering the limited depth sensitivity of the XPS test on the nanometer scale and the sample size being in the order of tens of microns, further analysis of the valence state of the Mo element in the H_xMoO_3 samples was conducted using Mo K-edge X-ray absorption near-edge structure (XANES) spectroscopy. The XANES spectra clearly indicated a shift towards lower energy absorption near the edge of H_xMoO_3 upon proton intercalation, compared to pristine $\alpha-MoO_3$, signifying a reduced oxidation state of Mo in the MoO_3 electrodes (Fig. 3d). To further validate the presence of the Mo^{3+} state in the $H_{2.15}MoO_3$ and $H_{2.83}MoO_3$ samples, we introduced a MoO_2 reference for comparison (Supplementary Fig. 12). The absorption near the edge of $H_{2.15}MoO_3$ exhibited a slight shift towards the lower energy region, with a portion of the edge curve remaining at higher energy compared to the MoO_2 reference. This observation confirmed the reduced oxidation state of Mo in the $H_{2.15}MoO_3$ electrode, consistent with the presence of the Mo^{3+} state. In contrast, the absorption near the edge of $H_{2.83}MoO_3$ displayed a significant shift towards the lower energy region, clearly distinguishing it from both the $H_{2.15}MoO_3$ electrode and the MoO_2 reference. This notable shift confirmed a substantial abundance of the Mo^{3+} state in $H_{2.83}MoO_3$, providing further support for the phenomenon of the third proton insertion. Overall, the XANES spectroscopy results not only addressed the limitations of XPS but also

provided compelling evidence for the reduced oxidation state of Mo and the presence of the Mo^{3+} state in the MoO_3 electrodes upon proton intercalation.

Fig. R1 Characterizations of the Mo valence states in MoO_3 electrodes during protons (de)intercalation process. **a** The Mo 3d XPS spectrums of MoO_3 electrodes performed at different discharge states. **b** The XANES spectra of H_xMoO_3 compared with Mo foil. **c** The XANES spectra of HMoO_3 compared with $\alpha\text{-MoO}_3$ and Mo foil references.

Reviewer #2, Comment #2

2. The authors compared the designed PSL electrolyte and 1 M H_3PO_4 electrolyte in Figure 1 b in terms of the voltage window. How about a concentrated H_3PO_4 with different concentrations?

Thank you for your comments. Fig. R2 illustrates that the PSL electrolyte demonstrates a lower hydrogen evolution current in comparison to the 1 M and 14.5 M phosphoric acid, primarily attributed to its water activity inhibition. Fig. R3 depicts extended peaks of OH groups and acid-acid hydrogen bonds around 2750 cm^{-1} and 1630 cm^{-1} , respectively, as well as a blue-shift of the ν P(OH)₃ peak around 930 cm^{-1} when compared to the 1 M and 14.5 M phosphoric acid. These spectral changes indicate the formation of a robust hydrogen-bonding network between the acid, water, and surfactant molecules, providing further confirmation of water activity inhibition within the PSL electrolyte.

Fig. R2 The electrochemical stability window of different H_3PO_4 and PSL electrolytes using linear sweep voltammetry (0.5 mV s^{-1}).

Fig. R3 Fourier transform infrared spectroscopy (FTIR) of different phosphoric acid (PA) electrolytes.

Reviewer #2, Comment #3

3. What would be the “effective” concentration of H_3PO_4 in water when the surfactant is added?

Thank you for your comments. We have calculated the concentration of H_3PO_4 in weight molar of 7.3 m. However, obtaining an accurate value for the "effective" concentration of H_3PO_4 in the PSL electrolyte poses challenges. This is primarily due to the presence of water molecules being immobilized within the liquid crystal structure of the PSL electrolyte through hydrogen bonding, making it difficult to quantify the amount of free water. From our analysis, it can be inferred that the "effective" concentration of H_3PO_4 in the PSL electrolyte maybe exceed 14.5 M but be lower than that of pure H_3PO_4 crystals.

Reviewer #2, Comment #4

4. The first protonation process is partially irreversible, where the lowest-potential plateau is not quite reversible. Do protons get trapped at low potentials? Or that is HER. Overall, the proton insertion/extraction coulombic efficiency is not very high for initial cycles. Why? And does the CE get improved over cycling?

Thank you for your valuable comments. In fact, we observed that protons become trapped after the initial redox peak in the first discharge process, as illustrated in Fig. R4 a. By the end of the first discharge cycle, approximately 2.83 moles of protons were inserted. However, during the subsequent deprotonation process, only 1.83 moles of protons were extracted, resulting in a lower coulombic efficiency. Nevertheless, it is

noteworthy that the trapped protons in HMoO_3 induced a semiconductor-to-metal transition, enabling a reversible discharge/charge process with favorable cycling performance after the initial cycle (Fig. R4 b).

At the same time, the cracking behavior was observed using focused ion beam (FIB) techniques for HR-SEM and HR-STEM tests after the first redox peak during the first discharge process, which cause the crystalline phase change (transformation from orthorhombic to monoclinic crystal structure).¹ As shown in Fig. R 5, the MoO_3 particles cracking behavior was observed and fractures inside the particles were also confirmed by FIB and HR-SEM (Fig. R 6 and R 7). For comparison, the lamella sample of pristine $\alpha\text{-MoO}_3$ was also prepared by FIB and no cracking features was observed, as shown in Fig. R 8. However, numerous linked cracks were observed in the HMoO_3 lamella sample, facilitating the water absorption and proton diffusion during the discharge/charge process (Fig. R 9).

According to this additional work and the suggestions from the reviewer, we have revised the manuscript and added these figures into Supplementary information.

Fig. R4 **a** The first CV curve of MoO₃ electrode at 1 mV s⁻¹ using the designed PSL electrolyte. **b** GCD curves of MoO₃ electrodes after the first cycle in PSL electrolyte.

Fig. R5 The SEM images of MoO₃ electrode after the initial redox peak in the first discharge process using PSL electrolyte.

Fig. R6 The SEM images of MoO₃ electrode after the initial redox peak in the first discharge process using PSL electrolyte. a-d Sectional view SEM image of MoO₃ electrode after the FIB cutting on the left side (1 μm cutting each from a to d).

Fig. R7 The SEM images of MoO₃ electrode after the initial redox peak in the first discharge process using PSL electrolyte. a-d Sectional view SEM image of MoO₃ electrode after the FIB cutting on the right side (1 μm cutting each from a to d).

Fig. R8 a to c STEM images of the FIB cut pristine α-MoO₃ particle with different magnifications viewed along the [001] direction. d to f STEM images of the FIB cut pristine α-MoO₃ particle with different magnifications viewed along the [010] direction.

Fig. R9 **a to c** STEM images of the FIB cut HMoO_3 particle with different magnifications viewed along the [001] direction. **d to f** STEM images of the FIB cut HMoO_3 particle with different magnifications viewed along the [010] direction.

Reviewer #2, Comment #5

5. For the storage mechanism of MoO_3 , it seems that the authors consider the storage of both “naked” protons as well as hydroniums, where the latter is used to explain the cracking behavior. Are all the inserted protons in the format of hydroniums? Or just partially.

Thank you for your comments. Actually, only part of the inserted protons is in the format of hydroniums. And the cracking behavior was observed after the initial redox peak in the first discharge process as we explained in question 4. The in-situ electrochemical quartz crystal microbalance (EQCM) measurements conducted by Ji and co-workers showed that water co-intercalation in the MoO_3 crystal or water adsorption on the MoO_3 particles surface during the proton (de)intercalation process, especially the one that exhibited a shoulder peak in the CV curve using bulk sizes α -

MoO₃ electrode material.². In this work, we also found two shoulder peaks in the CV curve collected from the MoO₃ electrode using the PSL electrolyte (Fig. R 10), and there should be hydroniums insertion during the first and third redox process and “naked” protons insertion in the second redox process.

Fig. R10 The first CV curve of MoO₃ electrode at 1 mV s⁻¹ using the designed PSL electrolyte.

Reviewer #2, Comment #6

6. From Figure 3f, it is not clear that the distance of Mo atoms is turned longer after proton insertion. Please point out the direction of the referred-to lattice.

Thank you for your comments. We have conducted additional work to address the concerns raised by the reviewer.

The atomic structure of the MoO₃ electrode after the initial cycle was investigated using a low-dose high-resolution scanning transmission electron microscopy (HR-STEM) and density functional theory (DFT) calculations. In Fig. R11, lamella samples of MoO₃ in different charge states were prepared using FIB techniques for HR-STEM tests. Fig. R11 a and b depict the HR-STEM images of pristine α -MoO₃, highlighting the distinct van der Waals gap (0.7 nm) observed between the zigzag arrays of

molybdenum atoms, which aligns very well with the crystal structure (0.693 nm). Also, the selected area electron diffraction (SAED) patterns in Fig. R11 a and b approved the orthorhombic symmetry of the pristine α -MoO₃.

In Fig. R11 c and d, the observed interlayer space expansion (from 0.7 nm to 0.74 nm) and structural distortion (transformation from orthorhombic to monoclinic crystal structure) clearly indicate the effects of proton intercalation. The corresponding HR-STEM images and SAED patterns in Fig. R11 c and d provide further evidence of the crystal structure distortion, specifically reflected in the change of the β angle from 90° to 93°.

According to this additional work and the suggestions from the reviewer, we have revised the manuscript and added these figure R11 into manuscript.

Fig. R11 Characterizations of the MoO_3 structure during protons (de)intercalation after the first cycle. **a** and **b** The crystal structure, HR-STEM, and SAED pattern (from left to right) of pristine $\alpha\text{-MoO}_3$ under different zoom axis (**a**: viewed along the [001] direction and **b**: viewed along the [010] direction). **c** and **d** The DFT simulated crystal structure, HR-STEM, and SAED pattern of HMoO_3 under different zoom axis (**c**: viewed along the [001] direction and **d**: viewed along the [010] direction).

Reviewer #2, Comment #7

7. Figure 6b, I don't think it is relevant to set the cutoff voltage of a battery to zero. The capacity below a voltage of 0.4 V is of very low quality, which should not be used to count the total capacity of a full cell.

Thank you for your comments. We agree that the capacity below 0.4 V is in low quality, because it has very little contribution to battery energy density. We set the discharge cut-off voltage to zero volts primarily to explore the limitations of the prototype cells, as most work on proton cells has been reported.^{3, 4} For application, it is not practical to discharge the battery to a low voltage (less than 0.4V). Moving forward, we will continue to focus on optimizing the proton electrolyte and do some research on the cathode electrode optimization to construct high-rate and high-energy aqueous energy storage devices.

Reviewer #2, Comment #8

8. The N/P ratio of the full cell should be given.

Thank you for your comments. The N/P ratio of the full cell is 1.1 and we have added this information to the method of the manuscript.

Reference list

- 1 Guo, H. C. et al. Two-Phase Electrochemical Proton Transport and Storage in α -MoO₃ for Proton Batteries. *Cell Rep. Phys. Sci.* **1**, 100225 (2020).
- 2 Jiang, H. et al. A High-Rate Aqueous Proton Battery Delivering Power Below -78 °C via an Unfrozen Phosphoric Acid. *Adv Energy Mater.* **10**, 2000968 (2020).
- 3 Wang, S. et al. Acid-in-Clay Electrolyte for Wide-Temperature-Range and Long-Cycle Proton Batteries. *Adv Mater.* **23**, 2202063 (2022).
- 4 Guo, Z. et al. An organic/inorganic electrode-based hydronium-ion battery. *Nat. Commun.* **11**, 959 (2020).

REVIEWERS' COMMENTS

Reviewer #1 (Remarks to the Author):

I appreciate the authors' efforts in carrying out additional characterizations, including HR-TEM and XAS, which make the underlying proton storage mechanism clearer and better justified. However, I am still not convinced that reporting such a proton-storage material or electrochemistry could significantly impact the aqueous battery field. Even though the authors claimed the three-proton insertion, only the two-proton process can be considered reversible for energy storage. From the device point of view, such an anode can hardly contribute to constructing batteries with high energy density. Thereby, I am sorry that I cannot support its publication in Nature Communications.

Reviewer #2 (Remarks to the Author):

The authors have addressed the comments of both reviewers properly. I think it can be accepted.

Response Letter to Reviewers

Manuscript Number: Nature Communications (NCOMMS-22-51746)

FULL TITLE: Discovery of a Three-proton Insertion Mechanism in α -Molybdenum Trioxide Leading to Enhanced Charge Storage Capacity

Reviewer Comments: Black

Authors' Responses: Blue

Revised Content: Red

REVIEWER COMMENTS

Reviewer #1 (Remarks to the Author):

I appreciate the authors' efforts in carrying out additional characterizations, including HR-TEM and XAS, which make the underlying proton storage mechanism clearer and better justified. However, I am still not convinced that reporting such a proton-storage material or electrochemistry could significantly impact the aqueous battery field. Even though the authors claimed the three-proton insertion, only the two-proton process can be considered reversible for energy storage. From the device point of view, such an anode can hardly contribute to constructing batteries with high energy density. Thereby, I am sorry that I cannot support its publication in Nature Communications.

Authors' Reply:

We greatly appreciate your thoughtful review of our manuscript, and we are grateful for your acknowledgment of the additional characterizations, such as HR-TEM and XAS, which we included to elucidate the proton storage mechanism and provide a more

robust justification. Also, we understand your concerns regarding the potential impact of our reported proton-storage material and its relevance to the aqueous battery field. We acknowledge the distinction you've made between the three-proton insertion and the more practically reversible two-proton process for energy storage. Your perspective on the implications of constructing batteries with high energy density is noted.

While we respect your viewpoint, it's important to clarify that our work aims to contribute not only to the immediate advancement of aqueous battery technologies but also to provide a comprehensive understanding of the proton storage mechanism in this specific material. We believe such insights could pave the way for future innovations and novel strategies in energy storage research.

Once again, thank you for your thoughtful evaluation.

Reviewer #2 (Remarks to the Author):

The authors have addressed the comments of both reviewers properly. I think it can be accepted.

Authors' Reply:

Thank you for your recognition of our work. We truly appreciate your time and consideration in reviewing our manuscript.